# LRBA deficiency impairs autophagy and contributes to enhanced antigen presentation and T-cell dysregulation

Elena Sindram [1,2,3], Marie-Celine Deau [1,19], Laure-Anne Ligeon [4,19], Pablo Sanchez-Martin [5], Sigrun Nestel [6], Sophie Jung [1,7], Stefanie Ruf [8], Pankaj Mishra [9], Michele Proietti [1,10,11], Stefan Günther [9], Kathrin Thedieck [8,12,13], Eleni Roussa [6,14], Angelika Rambold [1,15], Christian Münz [4], Claudine Kraft [5,16], Bodo Grimbacher [1,11,16,17,18 ✉] & Laura Gámez-Díaz [1,16 ✉]

## Abstract

**Reduced autophagy is associated with the aberrant humoral response observed in lipopolysaccharide-responsive beige-like anchor protein (LRBA) deficiency; however, the molecular mechanisms and their impact on T-cell responses remain poorly understood. We identify two novel LRBA interactors, phosphoinositide 3-kinase regulatory subunit 4 (PIK3R4) and FYVE And Coiled-Coil Domain Autophagy Adaptor 1 (FYCO1), which each play key roles in autophagy. PIK3R4 facilitates the production of phosphatidylinositol-3 phosphate (PI(3)P) that promotes autophagosome formation and autophagosome-lysosome fusion, whereas FYCO1 supports autophagosome movement. LRBA-knockout (KO) cells show impaired PI(3)P production, reduced autophagosome-lysosome fusion, accumulation of enlarged autophagosomes, and decreased cargo degradation. In line with the role of autophagy as a major degradation system for MHC-II loading and antigen presentation, we observe increased numbers of MHC class II and LC3 vesicles, along with enhanced antigen presentation in absence of LRBA, resulting in a higher production of proinflammatory cytokines from T cells in vitro. Our work suggests a novel biological role of LRBA controlling antigen presentation and T-cell responses by positively regulating autophagy, which may contribute to T-cell immune dysregulation observed in LRBA-deficient patients.**

**Keywords** Autophagy; FYCO1; Immune Dysregulation; LRBA; PIK3R4
**Subject Categories** Autophagy & Cell Death; Immunology; Molecular Biology of Disease

## Introduction

LRBA deficiency is a rare genetic immune disorder caused by deleterious biallelic mutations in *LRBA* (Lopez-Herrera et al, 2012). Clinically, it manifests with a broad spectrum of symptoms ranging from immunodeficiency to T-cell driven immune dysregulation, including inflammatory bowel disease (IBD) and autoimmune cytopenias (Alkhairy et al, 2016; Azizi et al, 2017a; Gamez-Diaz et al, 2016; Kostel Bal et al, 2017). Mechanistically, LRBA regulates the vesicular recycling of the cytotoxic T lymphocyte-associated protein 4 (CTLA-4) on regulatory T cells (Tregs), partly explaining the T-cell immune dysregulation upon loss of LRBA (Janman et al, 2021; Lo et al, 2015). Although the clinical pictures of LRBA deficiency and CTLA-4 insufficiency frequently overlap (Jamee et al, 2021; Lo et al, 2016), LRBA-deficient patients display an increased severity, poorer survival and earlier onset of symptoms than CTLA-4 insufficiency patients. These observations suggest that LRBA loss triggers additional disease mechanisms, beyond abnormal CTLA-4 trafficking.

We previously reported that B lymphocytes from LRBA-deficient patients exhibit defective autophagy leading to poor B cell survival (Lopez-Herrera et al, 2012). This defect pointed

[1]Institute for Immunodeficiency, Center for Chronic Immunodeficiency (CCI), Medical Center—University of Freiburg, Faculty of Medicine, University of Freiburg, Freiburg, Germany. [2]Spemann Graduate School of Biology and Medicine (SGBM), University of Freiburg, Freiburg, Germany. [3]Faculty of Biology, University of Freiburg, Freiburg, Germany. [4]Viral Immunobiology, Institute of Experimental Immunology, University of Zürich, Zürich, Switzerland. [5]Institute of Biochemistry and Molecular Biology, ZBMZ, Faculty of Medicine, University of Freiburg, Freiburg, Germany. [6]Institute of Anatomy and Cell Biology Department of Neuroanatomy, Faculty of Medicine, University of Freiburg, Freiburg, Germany. [7]Faculté de Chirurgie Dentaire, Université de Strasbourg—Pôle de Médecine et de Chirurgie Bucco-Dentaires, Hôpitaux Universitaires de Strasbourg, Strasbourg, France. [8]Laboratory of Pediatrics, Section Systems Medicine of Metabolism and Signaling, University of Groningen, University Medical Center Groningen, Groningen, The Netherlands. [9]Research group of Pharmaceutical Bioinformatics, Institute of Pharmaceutical Sciences, University of Freiburg, Freiburg, Germany. [10]Department of Rheumatology and Immunology, Hannover Medical School, Hannover, Germany. [11]RESIST—Cluster of Excellence 2155, Hanover Medical School, Satellite Center Freiburg, Freiburg, Germany. [12]Department for Neuroscience, School of Medicine and Health Sciences, Carl von Ossietzky University Oldenburg, Oldenburg, Germany. [13]Institute of Biochemistry and Center for Molecular Biosciences Innsbruck, University of Innsbruck, Innsbruck, Austria. [14]Institute of Anatomy and Cell Biology, Department of Molecular Embryology, Faculty of Medicine, University of Freiburg, Freiburg, Germany. [15]Laboratory for Structural Metabolism of Inflammation, Institute for Medical Biochemistry, Center for Molecular Biology of Inflammation, University of Münster, Münster, Germany. [16]CIBSS—Centre for Integrative Biological Signalling Studies, University of Freiburg, Freiburg, Germany. [17]Clinic of Rheumatology and Clinical Immunology, Center for Chronic Immunodeficiency (CCI), Medical Center—University of Freiburg, Faculty of Medicine, University of Freiburg, Freiburg, Germany. [18]DZFI—German Center for Infection Research, Satellite Center Freiburg, Freiburg, Germany. [19]These authors contributed equally: Marie-Celine Deau, Laure-Anne Ligeon. ✉E-mail: bodo.grimbacher@uniklinik-freiburg.de; laura.gamez@uniklinik-freiburg.de

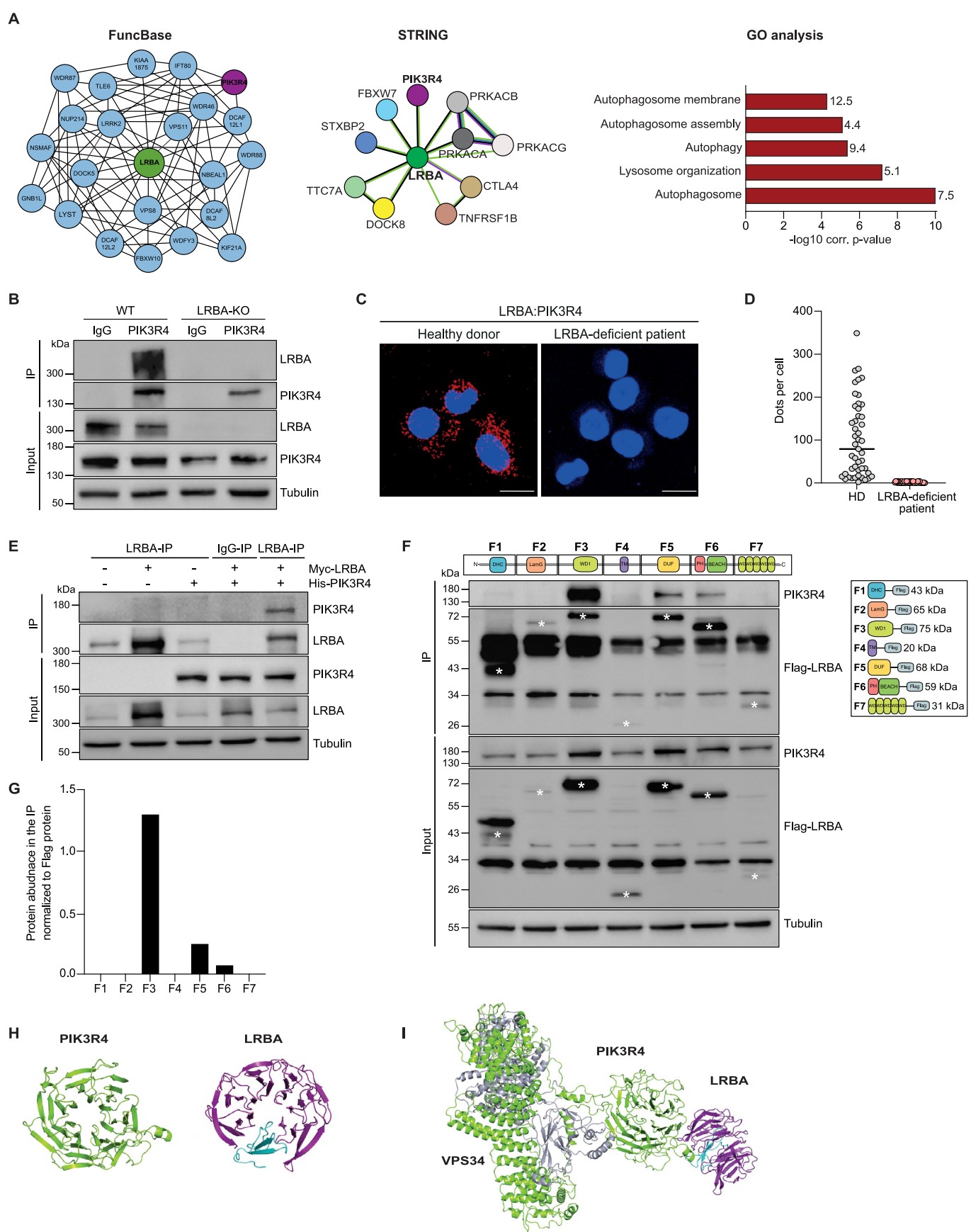

**Figure 1. LRBA interacts with PIK3R4 through its WD40 domain.**

(A) In silico identification of LRBA protein interaction network using FuncBase (left) and STRING version 12 (middle). In STRING, colored lines indicate evidence of text mining (green), co-occurrence (blue), co-expression (black) and experimental evidence (purple). GO enrichment analyses (right) of 31 potential interactors of LRBA for the GO domain "cellular compartment". The length of the bar represents the log10 Benjamini–Hochberg corrected P value. The numbers represent the percentage of associated genes for each biological process. (B) Co-IP analyses of endogenous LRBA and PIK3R4 interaction in WT and LRBA-KO HEK293T cells. PIK3R4 was immunoprecipitated with anti-PIK3R4. Western blot analysis of the immunoprecipitation and the input was performed with the indicated antibodies. Tubulin expression was used as an input control. (C) Representative fluorescent microscopy images of PLA signal in healthy donor (HD) and LRBA-deficient patient LCL cells. LRBA-PIK3R4 interaction is shown as a red signal and nuclear DAPI staining is shown in blue. Scale bar = 20 μm. (D) Dots (PLA signal) per cell were quantified in 50 cells per experiment from n = 2 biological replicates of HD (grey) and LRBA-deficient (red) LCL cells using Duolink Image tool. (E) HEK293T WT cells were co-transfected with full-length Myc-LRBA plasmid and/or full-length Myc-VPS34/His-PIK3R4 plasmid (VPS34 blots are shown in Fig. EV1C). Immunoprecipitation was performed with anti-LRBA or anti-IgG and immunoblotted for PIK3R4. (F) HEK293T WT cells were co-transfected with His-tagged full-length PIK3R4 and Flag-tagged LRBA protein domains (fragments 1 to 7). Flag-LRBA was immunoprecipitated with anti-Flag, and western blots were probed with anti-PIK3R4. Schematic representation of plasmids containing the different LRBA protein domains is shown on the top and right. Asterisks in the western blots indicate the LRBA protein domains. (G) Ratio of PIK3R4/LRBA fragment detected by co-IP was calculated for n = 1 biological replicates. (H) Models of the WD40 domains of PIK3R4 (light green) and LRBA (purple) using the model server WDSPdb 2.0. The inserted repeat propeller of LRBA is shown in cyan. (I) Model of the complex of LRBA (purple) with PIK3R4 (light green)/PIK3C3 (VPS34, silver). The inserted repeat propeller of LRBA-WD40 is shown in cyan. Source data are available online for this figure.

towards impaired fusion between autophagosomes and lysosomes and/or non-functional lysosomes; however, the mechanism remained unclear. Autophagy occurs in all eukaryotic cells, engulfing cytoplasmic material, including aberrant proteins or intracellular pathogens, in vesicles called autophagosomes, which subsequently fuse with lysosomes for cargo degradation (Mizushima, 2007). Beyond maintaining cellular homeostasis, autophagy regulates immune recognition and responsiveness. For instance, autophagy supports antigen processing and MHC-II-mediated antigen presentation in professional and non-professional antigen presenting cells (Cui et al, 2019; Dikic and Elazar, 2018; Schmid et al, 2007). Specifically, autophagosomes fuse with multivesicular MHC-II-loading compartments (MIIC), leading to the degradation of intracellular proteins and pathogens into peptides that can be loaded onto MHC-II molecules (Schmid et al, 2007). Whether aberrant autophagy arising from LRBA loss impacts antigen presentation is unknown. Moreover, IBD, a prominent symptom of LRBA deficiency characterized by chronic inflammation, is also linked to abnormal autophagy (Hampe et al, 2007; Shao et al, 2021).

In this study, we demonstrate that LRBA physically interacts with proteins that are essential for autophagy, including phosphoinositide-3-kinase regulatory subunit 4 (PIK3R4, also known as VPS15) and UV radiation resistance associated (UVRAG), two components of a phosphatidylinositol 3-kinase III (PI3K-III) complex. We show that LRBA loss diminished PI(3)P production, altered subcellular localization of PI(3)P-binding proteins, inhibited autophagosome-lysosome fusion, and led to autophagosome accumulation and abnormal autophagy flux in multiple cell types. In addition, we show that LRBA interacts with FYVE and Coiled-Coil Domain Autophagy Adaptor 1 (FYCO1) (Nieto-Torres et al, 2021), which links autophagosomes to microtubule plus-end-directed molecular motors, and regulates autophagosome movement (Pankiv et al, 2010; Pankiv and Johansen, 2010). Importantly, we provide evidence that LRBA regulates MHC-II-restricted antigen presentation via autophagy, thereby regulating stimulation of CD4+ T cells and curbing the release of T-cell proinflammatory cytokines. This novel function of LRBA in regulating antigen presentation may contribute to the T-cell immune dysregulation observed in LRBA-deficient patients.

## Results

### LRBA interacts with components of a phosphatidylinositol 3-kinase III (PI3K-III) complex

Using computational predictions based on STRING (v12) and FuncBase databases (Beaver et al, 2010; Szklarczyk et al, 2017), we identified 31 potential novel LRBA interactors that were enriched in autophagy and vesicle-trafficking events according to Gene Ontology (GO) term enrichment analyses (Fig. 1A; Table 1). PIK3R4, a component of the PI3K-III core complex that controls autophagy and endocytosis, was predicted to interact with LRBA in both databases. To investigate this potential interaction, we performed co-immunoprecipitation (co-IP) and proximity ligation assays (PLA). Endogenous LRBA interacted with PIK3R4 by co-IP in HEK293T cells (Fig. 1B) and by PLA in lymphoblastoid B cells (LCL) from a healthy donor (Fig. 1C,D). Similarly, we detected a physical interaction between overexpressed Myc-tagged LRBA and His-tagged PIK3R4 in HEK293T cells (Fig. 1E).

LRBA contains several different domains that are involved in protein-protein interactions (Fig. 1F, top) (Cullinane et al, 2013; Martinez Jaramillo and Trujillo-Vargas, 2018). To determine the domains of LRBA that interact with PIK3R4, we co-expressed Flag-tagged fragments of LRBA (F1 to F7) with His-tagged full-length PIK3R4 in HEK293T cells (Fig. 1F, top and right panel). By co-IP, we detected a strong interaction between His-PIK3R4 and fragment 3 (F3) of LRBA, which contains the central WD40 domain of LRBA (Fig. 1F,G).

To explore the LRBA-PIK3R4 interaction further, we used High Ambiguity Driven protein-protein DOCKing (HADDOCK), an integrative platform for the modeling of biomolecular complexes (Dominguez et al, 2003; van Zundert et al, 2016). Homology modelling of the individual WD40 domains of LRBA and PIK3R4 predicted the well-established β-propeller structure. The WD40 domain of PIK3R4 was predicted to form a single helix on the surface, which served as an interacting constraint for the docking protocol (Fig. 1H). In the best model, the exposed α-helix of PIK3R4 interacts with the upper surface of the LRBA β-propeller via hydrophobic and charged interactions (Fig. 1I).

Beyond PIK3R4, the PI3K-III core complex contains VPS34 and Beclin-1 (McKnight et al, 2014). This core complex interacts with

**Table 1. Predicted LRBA interactors by computational predictions.**

| N° | Gen | Protein code | Protein | Function | Database |
|---|---|---|---|---|---|
| 1 | CTLA4 | P16410 | Cytotoxic T-lymphocyte associated protein | Inhibitory receptor acting as a major negative regulator of T-cell responses | STRING |
| 2 | DCAFI2L1 | Q5VU92 | DDB1- and CUL4-associated factor 12-like protein 1 | Unknown | FuncBase |
| 3 | DCAFI2L2 | Q5VW00 | DDB1- and CUL4-associated factor 12-like protein 2 | Unknown | FuncBase |
| 4 | DCAF8L2 | P0C7V8 | DDB1- and CUL4-associated factor 8-like protein 2 | Unknown | FuncBase |
| 5 | DOCK5 | Q9H7D0 | Dedicator of cytokinesis protein 5 | Epithelial cell migration | FuncBase |
| 6 | DOCK8 | Q8NF50 | Dedicator of cytokinesis protein 8 | Migration | STRING |
| 7 | FBXW7 | Q969H0 | F-Box and WD repeat domain containing 7 | Ubiquitination and subsequent proteasomal degradation of target proteins | STRING |
| 8 | FBXW10 | Q5XX13 | F-box/WD repeat-containing protein 10 | Ubiquitination and subsequent proteasomal degradation of target proteins | FuncBase |
| 9 | GNB1L | Q9BYB4 | Guanine nucleotide-binding protein subunit beta-like protein 1 | Intracellular signal transduction | FuncBase |
| 10 | IFT80 | Q9P2H3 | Intraflagellar transport protein 80 homolog | Development and maintenance of motile and sensory cilia | FuncBase |
| 11 | KIAA1875 | A6NE52 | WD repeat-containing protein 97 | Generation of catalytic spliceosome | FuncBase |
| 12 | KIF21A | Q7Z4S6 | Kinesin-like protein KIF21A | Microtubule-based movement | FuncBase |
| 13 | **LRRK2** | **Q5S007** | **Leucine-rich repeat serine/threonine-protein kinase** | **Autophagy through the CaMKK/AMPK** | **FuncBase** |
| 14 | LYST | Q99698 | Lysosomal-trafficking regulator | Sorting endosomal resident proteins into late multivesicular endosomes | FuncBase |
| 15 | NBEAL1 | Q6Z530 | Neurobeachin-like protein 1 | Phospholipid binding | FuncBase |
| 16 | NSMAF | Q92636 | Protein FAN | Ceramide metabolic process | FuncBase |
| 17 | NUP214 | P35658 | Nuclear pore complex protein Nup214 | Nuclear export signal receptor activity | FuncBase |
| 18 | **PIK3R4** | **Q99570** | **Phosphoinositide 3-kinase regulatory subunit 4** | **Macroautophagy and late endosome to vacuole transport** | **FuncBase STRING** |
| 19 | PRKACA | P17612 | Protein kinase cAMP-activated catalytic subunit alpha | Phosphorylation of substrates in the cytoplasm and the nucleus | STRING |
| 20 | PRKACB | P22694 | Protein kinase cAMP-activated catalytic subunit beta | Phosphorylation of substrates in the cytoplasm and the nucleus | STRING |
| 21 | PRKACG | P22612 | Protein kinase cAMP-activated catalytic subunit gamma | Phosphorylation of substrates in the cytoplasm and the nucleus | STRING |
| 22 | STXBP2 | Q15833 | Syntaxin binding protein 2 | Intracellular vesicle trafficking and vesicle fusion with membranes | STRING |
| 23 | TLE6 | Q9H808 | Transducin-like enhancer protein 6 | Positive regulation of neuron differentiation | FuncBase |
| 24 | TNFRSF1B | P20333 | TNF receptor superfamily member 1B | Calcineurin- dependent activation of NF-AT, NF-kappa-B and AP-1 | STRING |
| 25 | TTC7A | Q9ULT0 | Tetratricopeptide repeat domain 7A | Regulator of phosphatidylinositol 4-phosphate (PtdIns(4)P) synthesis | STRING |
| 26 | **VPS8** | **Q8N3P4** | **Vacuolar protein sorting-associated protein 8 homolog** | **Endosomal vesicle fusion** | **FuncBase** |
| 27 | **VPS11** | **Q9H270** | **Vacuolar protein sorting-associated protein 11 homolog** | **Vesicle-mediated protein trafficking** | **FuncBase** |
| 28 | WDR88 | Q6ZMY6 | WD repeat-containing protein 88 | Unknown | FuncBase |
| 29 | WDFY3 | Q8IZQ1 | WD repeat- and FYVE-containing protein 3 | Aggrephagy | FuncBase |
| 30 | WDR46 | O15213 | WD repeat-containing protein 46 | Scaffold component of the nucleolar structure | FuncBase |
| 31 | WDR87 | Q6ZQQ6 | WD repeat-containing protein 87 | Unknown | FuncBase |

List of the 31 potential LRBA interactors identified by FuncBase and/or STRING. Proteins are listed alphabetically. Proteins highlighted in bold were selected for validation by co-IP or PLA shown in Fig. 1 or EV1.

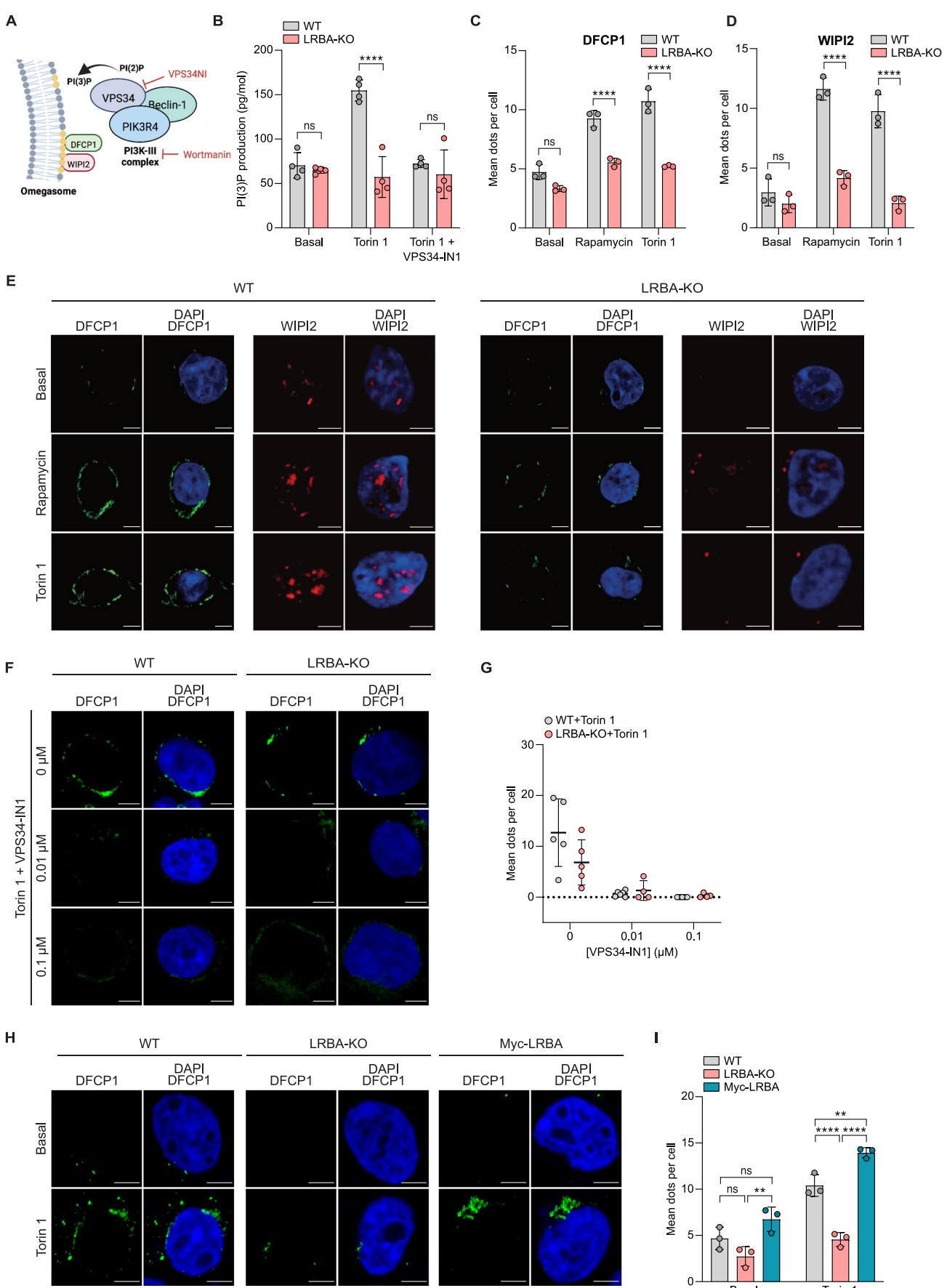

◀ Figure 2. LRBA deficiency reduces PI(3)P production and WIPI2 and DFCP1 punctae.

(A) Schematic representation of PI3K-III complex activity and the initiation of autophagosome formation (omegasome). (B) PI(3)P levels were measured in lipid extracts of WT (grey) and LRBA-KO (red) HEK293T cells under basal conditions, or upon treatment with 333 nM of Torin 1 alone, or in combination with 1 nM of VPS34-IN1 (PI3K-III complex inhibitor) for 4 h. Each dot represents technical replicates ($n = 2$) from one experiment, while bars represent the mean ± SD of $n = 4$ independent biological experiments. (C, D) Quantification of (C) DFCP1 and (D) WIPI2 signal in WT (grey) and LRBA-KO (red) HEK293T cells immunostained with either (C) anti-DFCP-1 or (D) anti-WIPI2 antibody after incubation for 1 h with either control medium, 50 μM Rapamycin, or 333 nM of Torin 1. Each dot represents the mean of one experiment while the bars represent the mean ± SD of $n = 3$ independent biological experiments. Total cells analyzed in (C, D) were: basal conditions, WT = 27/59, KO = 21/35; Rapamycin, WT = 44/70, KO = 49/91 and Torin 1, WT = 52/51, KO = 49/41. (E) Representative confocal microscopy images of WT (left) and LRBA-KO (right) HEK293T cells immunostained with anti-DFCP1 (green) or anti-WIPI2 (red) under the indicated conditions. Scale bar = 5 μm. (F) Representative confocal microscopy images (Scale bar = 5 μm) and (G) quantification of DFCP1 signal in WT (grey) and LRBA-KO (red) HEK293T cells immunostained with anti-DFCP1 after treatment with the indicated concentrations of VPS34-IN1 for 1 h followed by 1 h incubation with 333 nM Torin 1. Each dot represents one image analyzed and the mean ± SD was calculated from n = 2 independent biological experiments. Total cells analyzed in WT: 0 μM = 46, 0.01 μM = 42, 0.1 μM = 42 and in KO: 0 μM = 29, 0.01 μM = 110, 0.1 μM = 94. Scale bar = 5 μm. (H, I) LRBA reconstitution restores DFCP1 recruitment. (H) Representative confocal microscopy images and (I) quantification of DFCP-1 expression in WT (grey), LRBA-KO (red) and Myc-LRBA (teal) HEK293T cells after stimulation for 1 h with control medium or 333 nM Torin 1. Each dot represents the mean of one experiment while bars represent the mean ± SD of $n = 3$ independent biological replicates. Total cells analyzed in WT (basal/Torin): 158/244 cells; LRBA-KO: 269/294 cells and Myc-LRBA: 152/273. Statistical analyses of (B–D) was performed using a two-way ANOVA with Bonferroni's multiple comparisons test and for (I) a two-way ANOVA with Tukey's multiple comparisons test, **$P < 0.01$ ((I): $P = 0.0014$, Basal LRBA-KO vs Myc-LRBA), ((I): $P = 0.0037$, Torin1 WT vs Myc-LRBA), ****$P < 0.0001$. Source data are available online for this figure.

various adaptor proteins, such as ATG14L and UVRAG (Ohashi et al, 2019; Yan et al, 2009). Our modelling predicted that PIK3R4 could indirectly bridge LRBA to VPS34 and, in turn, the entire PI3K-III core complex (Fig. 1I). To test this, we performed co-IP experiments and found that Myc-LRBA interacts with GFP-UVRAG in HEK293T cells (Fig. EV1A), although we did not detect physical interactions between LRBA and VPS34, Beclin-1, or ATG14L (Fig. EV1B–D). In addition, endogenous LRBA did not interact with the PI3K family member PI3K-delta (Fig. EV1E), an essential protein for immune responses (Nunes-Santos et al, 2019). Altogether, these results suggest that LRBA interacts with PIK3R4 and UVRAG, which positively regulate the PI3K-III complex (Abrahamsen et al, 2012; Liang et al, 2008).

## LRBA deficiency reduces PI3K-III activity and WIPI2 and DFCP1 punctae

The PI3K-III complex promotes the initiation and expansion of autophagosomes by generating PI(3)P, which is recognized by WIPI2 (WD-repeat protein Interacting with PhosphoInositides 2) and DFCP1 (Double FYVE domain-containing protein 1) proteins (Axe et al, 2008; Backer, 2016; Funderburk et al, 2010; Nascimbeni et al, 2017) (Fig. 2A). Given our observation that LRBA interacts with PIK3R4, we considered that LRBA deficiency would affect autophagy by altering PI3K-III complex activity. To test this, we induced autophagy by inhibiting mTORC1 with Torin 1 in wild-type (WT) and LRBA-KO HEK293T cells (Kim and Guan, 2015). Autophagy induction led to increased levels of PI(3)P in lipid extracts from WT but not from LRBA-KO HEK293T cells (Fig. 2B). As a positive control for PI3K-III complex inhibition, we used VPS34-IN1 to block VPS34 activity. This treatment effectively prevented the Torin 1-induced increase in PI(3)P levels in WT HEK293T cells, mimicking the PI3-III complex inhibition observed in LRBA-KO cells (Fig. 2B). Overall, these data suggest that LRBA deficiency reduces PI(3)P production during autophagy.

PI(3)P supports the translocation of the PI(3)P-binding proteins DFCP1 and WIPI2 into punctate compartments during autophagosome formation (Fig. 2A) (Axe et al, 2008; Palamiuc et al, 2020). WIPI2 and DFCP-1 accumulated in puncta in WT HEK293T cells after inducing autophagy by Torin 1 or another mTORC1 inhibitor,

Rapamycin (Fig. 2C–E), as expected. In contrast, the number of DFCP1 and WIPI2 puncta were reduced in LRBA-KO cells compared to WT cells treated with mTORC1 inhibitors (Fig. 2C–E). VPS34-IN1 treatment dramatically reduced the number of DFCP1 puncta in WT HEK293T cells treated with Torin 1, consistent with PI3K-III inhibition, lower PI(3)P levels and reduced autophagy (Fig. 2F,G) (Jaber et al, 2012). VPS34-IN1 treatment had a smaller impact on DFCP1 puncta in LRBA-KO HEK293T cells treated with Torin 1. Importantly, DFCP1 puncta were restored in LRBA-KO cells transfected with Myc-LRBA (Figs. 2H,I and EV2A). Notably, expression levels of PIK3R4, VPS34, WIPI2 and DFCP1 were similar in WT and LRBA-KO HEK293T cells, irrespective of cells genotype or Torin 1 treatment (Fig. EV2B–F), confirming that loss of LRBA affects PIK3R4 activity and WIPI2/DFCP1 punctae but not protein expression.

In addition, we observed severe reduction of WIPI2/DFCP1 punctae in LRBA-KO cells treated with mTORC1 inhibitors than under starvation (Figs. 2E and EV2G–I), suggesting that LRBA-KO cells were less sensitive to mTOR inhibition than WT cells. Indeed, phosphorylation of mTOR targets p70-S6K1 and S6RP were elevated in cells lacking LRBA compared to controls (Fig. EV3A–D), suggesting that mTOR activity is higher in the absence of LRBA. Reconstitution of LRBA-KO HEK293T cells with Myc- LRBA restored S6RP phosphorylation to WT cells levels (Fig. EV3E–G).

Overall, these data suggest that LRBA deficiency reduces PI3K-III activity, leading to decreased PI(3)P levels and impaired recruitment of PI(3)P-binding proteins WIPI2 and DFCP1, potentially compromising autophagy flux.

## LRBA-deficient cells have impaired autophagosome-lysosome fusion and p62 degradation

To determine whether LRBA is required for proper autophagic flux, we used HEK293T cells stably expressing the mCherry-GFP-LC3 tandem vector. This dual-tag system discerns early autophagosomes (green and red signal) from mature autolysosomes (red signal) (Lopez et al, 2018). Therefore, a high ratio of red foci/green foci reflects high autophagic flux. WT HEK293T cells expressing mCherry-GFP-LC3 and treated with Torin 1 showed a high ratio of red foci/green foci, which was reduced by inhibiting autophagic flux

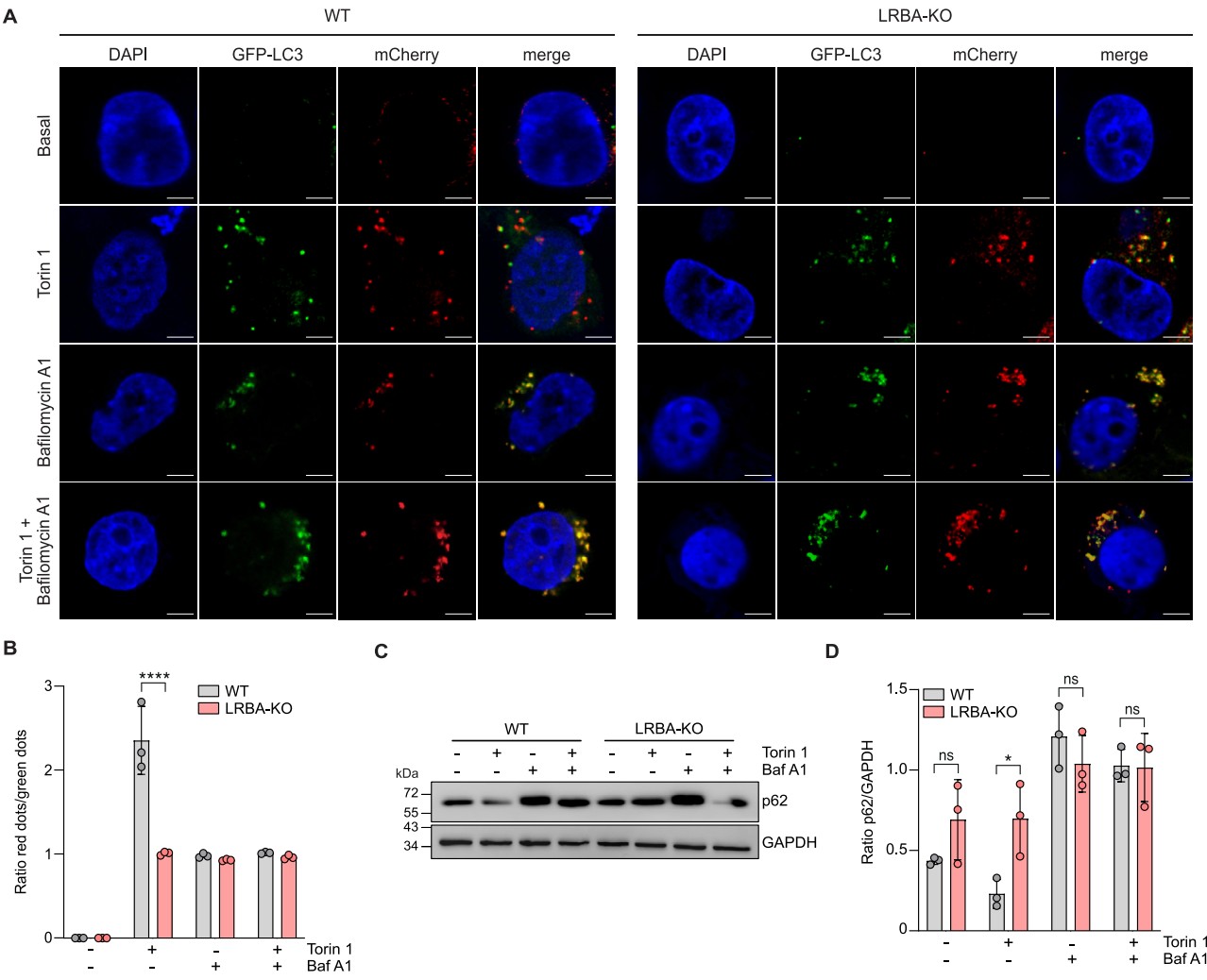

**Figure 3. LRBA-deficient cells have impaired autophagosome-lysosome fusion and p62 degradation.**

(A) Representative confocal microscopy images of WT and LRBA-KO HEK293T cells stably transduced with mCherry-GFP-LC3 tandem vector after 4 h treatment with 333 nM Torin 1 alone, 200 nM Bafilomycin A1 alone, or in a combination of both. Scale bar = 5 μm. (B) Ratio red/green dots quantification from (A). Each dot represents the mean of one experiment while bars represent the mean ± SD of $n = 3$ independent biological replicates. Total cells analyzed in WT: Torin 1 = 87, Bafilomycin A1 = 86, Torin 1+Bafilomycin A1 = 86 cells; LRBA-KO: Torin 1 = 81, Bafilomycin A1 = 93, Torin 1+Bafilomycin A1 = 84 cells. (C) Representative immunoblot analyses of p62 expression in WT and LRBA-KO HEK293T cells at basal conditions or after 4 h stimulation with 333 nM Torin 1 alone, with 200 nM Bafilomycin A1 alone, or with a combination of both. GAPDH was used as housekeeping protein. (D) Densitometry analyses of p62 immunoblots in WT (grey) and LRBA-KO (red) HEK293T cells. Each dot represents the densitometry analysis of one blot while bars represent the mean ± SD from $n = 3$ independent biological replicates. Statistical analyses of (B, C) was performed using a two-way ANOVA with Bonferroni's multiple comparisons test *$P < 0.05$ ((D): $P = 0.0193$), ****$P < 0.0001$. Source data are available online for this figure.

with Bafilomycin A1 treatment (Fig. 3A,B). In contrast, LRBA-KO HEK293T cells stimulated with Torin 1 displayed a low red foci/ green foci ratio, which was unaffected by inhibiting autophagic flux with Bafilomycin A1 (Fig. 3A,B). Additionally, we monitored the levels of p62, a cargo protein that is degraded during autophagy (Lopez et al, 2018). We observed reduced levels of p62 in WT HEK293T cells following Torin 1 treatment, which was effectively blocked by treatment with Bafilomycin A1, as expected (Fig. 3C,D). In contrast, p62 levels did not change in LRBA-KO HEK293T cells treated with Torin-1 (Fig. 3C,D), and were rescued by LRBA reconstitution with Myc-LRBA (Fig. EV4A,B).

Abnormal autophagy flux was also observed in immune cells upon autophagy induction. Specifically, we observed reduced p62 degradation following Rapamycin treatment in LRBA-KO Ramos B cells (Fig. EV4C,D), and a lower LC3-II/GAPDH ratio after LPS stimulation in B cells from *Lrba*−/− mice, compared to their wild-type counterparts (Fig. EV4E,F). Similar results were obtained in human shLRBA HeLa cells upon autophagy induction with MG132 (Fig. EV4G–J) (Lagos et al, 2022). MG132 inhibits proteasome activity, while LPS induces B cell differentiation and enhances immunoglobulin production. Both, MG132 and LPS stimulation result in protein accumulation and cellular stress, which

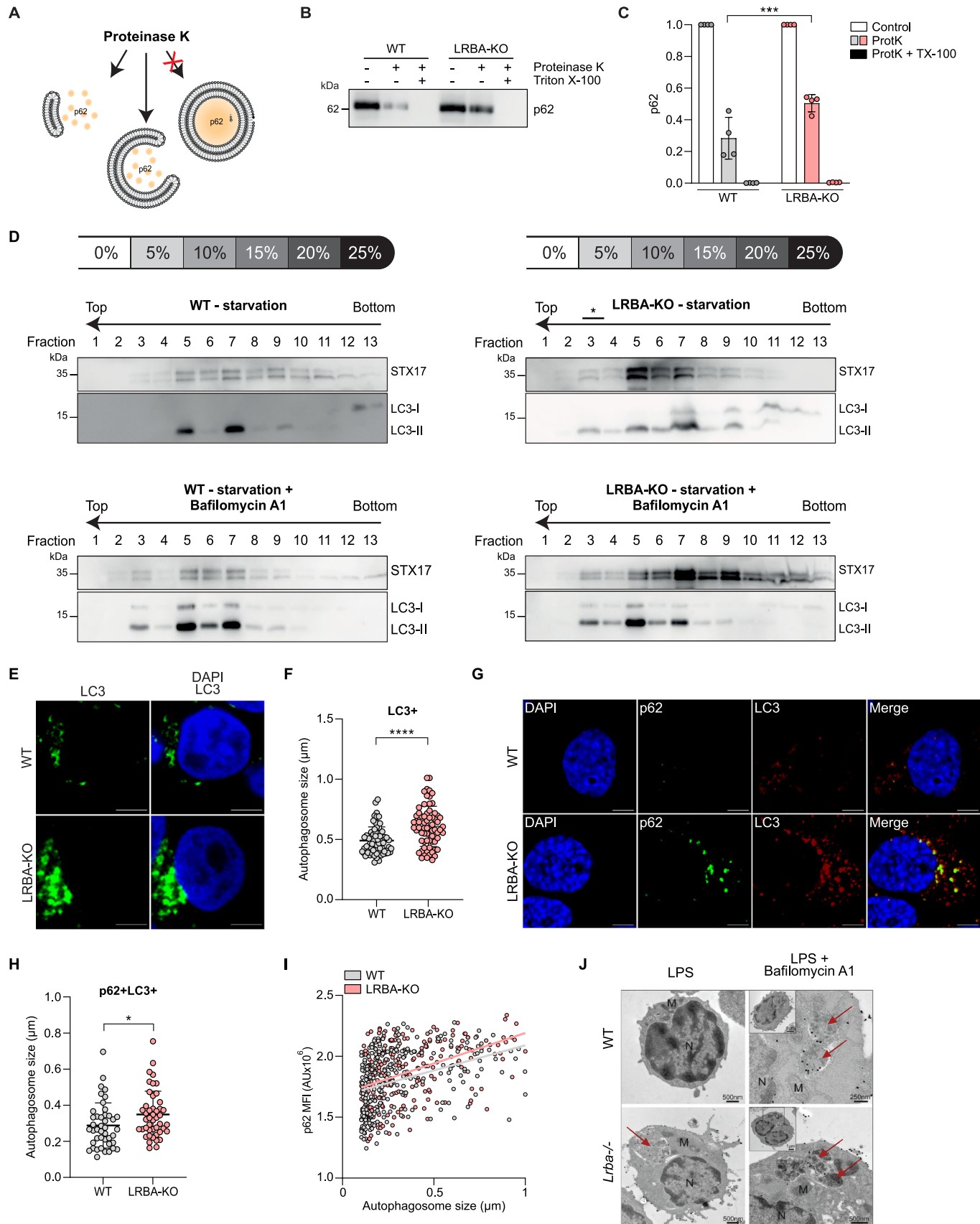

Figure 4.   Loss of LRBA leads to abnormal accumulation of enlarged, sealed and unfused autophagosomes.

(A) Schematic illustration of the protease protection assay. Cell homogenates are treated with proteinase K resulting in the degradation of exposed proteins but not membrane-enclosed proteins. (B) WT and LRBA-KO HEK293T cells were grown in EBSS medium with 300 nM Torin 1 for 2 h. After lysis, cell membranes were subjected to proteinase K and Triton X-100 treatment as indicated and analyzed by anti-p62 western blotting. (C) Quantification of (B) in WT (grey) and LRBA-KO (red) HEK293T cells. Each dot represents the densitometry analysis of one blot while bars represent the mean ± SEM from n = 4 independent biological replicates. (D) WT and LRBA-KO HEK293T cells were grown in EBSS medium with 300 nM Torin 1 for 3 h with or without 200 nM Bafilomycin A1, and cell homogenates were subjected to OptiPrep flotation analyses. The asterisk indicates the accumulation of autophagosomes in LRBA-KO homogenates. (E) Representative confocal microscopy images of LC3 signal in WT and LRBA-KO HEK293T cells stained with anti-LC3 after Bafilomycin A1 stimulation for 1 h. Scale bar = 5 µm. (F) Scatter plots showing the autophagosome size (in µm) of (E) in WT (grey) and LRBA-KO (red) HEK293T cells. Each dot represents one autophagosome from n = 3 independent biological replicates (total n = 68 autophagosomes), mean ± SD shown by the black line and error bars. (G) Representative confocal microscopy images of WT and LRBA-KO HaCat cells stimulated overnight with IFN-γ and treated with 20 µM chloroquine for 6 h. Fixed cells were stained for p62 (green), LC3 (red) and DAPI (blue) for the nuclei. Scale bar = 5 µm. (H) Scatter plots showing the autophagosome size (in µm) of (G) in WT (grey) and LRBA-KO (red) HaCat cells. Each dot represents the average size of all autophagosomes from one cell from n = 3 independent biological replicates with a total of at least 43 cells analyzed per condition, mean ± SD shown by the black line and error bar. (I) Correlation of autophagosome size with p62 MFI in WT (grey) and LRBA-KO (red) HaCat cells. Each dot represents one autophagosome from n = 3 independent biological replicates with a total of n = 262 (WT) and n = 225 (LRBA-KO) autophagosomes analyzed. (J) Electron microscopy images on LPS-stimulated B cells from WT and $Lrba^{-/-}$ mice using immunogold labeling for LC3. Arrows point to autophagosomes, N: nucleus, M: mitochondria. Images represent higher magnification of the inset's black-boxed area. A total of three mice per genotype were analyzed. Scale bars = 500 nm for LPS WT and $Lrba^{-/-}$ and for $Lrba^{-/-}$LPS+Bafilomycin A1, 250 nm for WT LPS+Bafilomycin A1. Statistical analyses of (C) was performed using a two-way ANOVA with Bonferroni's multiple comparisons test, and for (F, H) a unpaired Welch's t test, *P < 0.05 ((H): P = 0.0275), ***P < 0.001 ((C): P = 0.001), ****P < 0.0001. Source data are available online for this figure.

subsequently activate autophagy (Ding et al, 2007; He et al, 2021; Pengo et al, 2013).

Overall, these data set suggest that autophagic flux and autophagosome-lysosome fusion is reduced in LRBA-deficient cells, leading to an accumulation of autophagosomes.

## Loss of LRBA leads to accumulation of large autophagosomes

To test whether the autophagosomes that accumulated in LRBA-KO cells are mature and sealed, we monitored the sensitivity of p62 to proteinase K; p62 is protected from degradation when it is present in sealed autophagosomes (Abada et al, 2017) (Fig. 4A). We performed this protease protection assay on WT and LRBA-KO HEK293T cells treated with Torin 1. Interestingly, LRBA deficiency rendered p62 less sensitive to proteinase K treatment (Fig. 4B,C). These data suggest that autophagosomes in LRBA-deficient cells are sealed and accumulating cargo.

Additionally, we performed membrane flotation assays using OptiPrep gradients on starved HEK293T cells, (Matsui et al, 2018) and examined the gradients by detecting LC3-II on western blots. Autophagosomes typically float at 5% OptiPrep, whereas autophagic precursor membranes, endoplasmic reticulum (ER) and lysosomes float at higher densities. In starved LRBA-KO HEK293T cells, LC3-II was detected not only at the 5% and 7% fractions as in WT cells, but also in the 3% and 4% fractions, indicating an accumulation of enlarged autophagosomes (Fig. 4D). WT cells treated with Bafilomycin A1 also accumulated larger autophagosomes in the 3% and 4% fractions (Fig. 4D), phenocopying loss of LRBA. We obtained similar results by monitoring the autophagosomal protein STX17 (Fig. 4D). Enlarged and accumulating autophagosomes were also observed in LRBA-KO HEK293T and shLRBA HeLa cells versus their control cells by detecting LC3 signal (Figs. 4E,F and EV5A,B), and in LRBA-KO HaCat cells by LC3+p62+ co-staining (Fig. 4G,H). A positive correlation between autophagosome size and cargo content was observed, indicating that larger autophagosomes carry more cargo (Fig. 4I). Notably, LRBA-KO HaCat cells showed an overall higher p62 cargo compared to WT cells (Fig. EV5C). Although co-localization analysis for p62+LC3+ signals was similar between

LRBA-KO and WT HaCat cells, we observed an increased accumulation of these vesicles in LRBA-deficient cells, supporting abnormal autophagosome-lysosome fusion (Fig. EV5D,E). In line with these results, LC3 immunogold labelling and electron microscopy (EM) also revealed larger autophagosomes in LPS-stimulated B cells from $Lrba^{-/-}$ mice versus WT mice (Fig. 4J).

These data indicate that in the absence of LRBA, autophagosomes fail to fuse with lysosomes leading to the accumulation of large cargo-containing autophagosomes.

## LRBA binds to FYCO1 facilitating proper autophagosome movement and lysosome positioning

In addition to the interaction between LRBA and regulators of the PI3K-III complex that we discovered here, LRBA was predicted to interact with FYCO1 in previous studies (Behrends et al, 2010). FYCO1 supports autophagosome transport and maturation by interacting with kinesin motor proteins and with autophagosome components (Mackeh et al, 2013; Pankiv et al, 2010; Pankiv and Johansen, 2010). We validated the potential interaction of LRBA with FYCO1 by co-IP in HEK293T cells co-transfected with GFP-tagged FYCO1 and Myc-tagged LRBA (Fig. 5A). In addition, we identified three fragments of LRBA that are sufficient to interact with FYCO1 (Fig. 5B,C). Notably, FYCO1 protein levels remain unaltered in LRBA-KO HEK293T compared to WT HEK293T cells, both before and after stimulation with Torin 1 (Fig. EV5F,G), suggesting that loss of LRBA does not affect FYCO1 stability.

To investigate whether LRBA deficiency affects autophagosome transport, we performed live-microscopy of HeLa cells expressing GFP-LC3 treated with Bafilomycin A1. We found that autophagosomes have a reduced velocity in shLBRA HeLa cells compared to shControl HeLa cells (Fig. 5D,E; Movies EV1 and EV2). Velocity was directly proportional to the autophagosome size (Fig. 5F).

Interestingly, LysoTracker Red staining revealed that lysosomal positioning was also altered in shLRBA HeLa cells compared to shControl cells (Fig. 5G). However, the levels and localization of Transcription Factor EB (TFEB), the master regulator of lysosomal biogenesis, were unaltered in shLRBA HeLa cells (Fig. 5H), suggesting that lysosome biogenesis is unaffected. EM analysis of

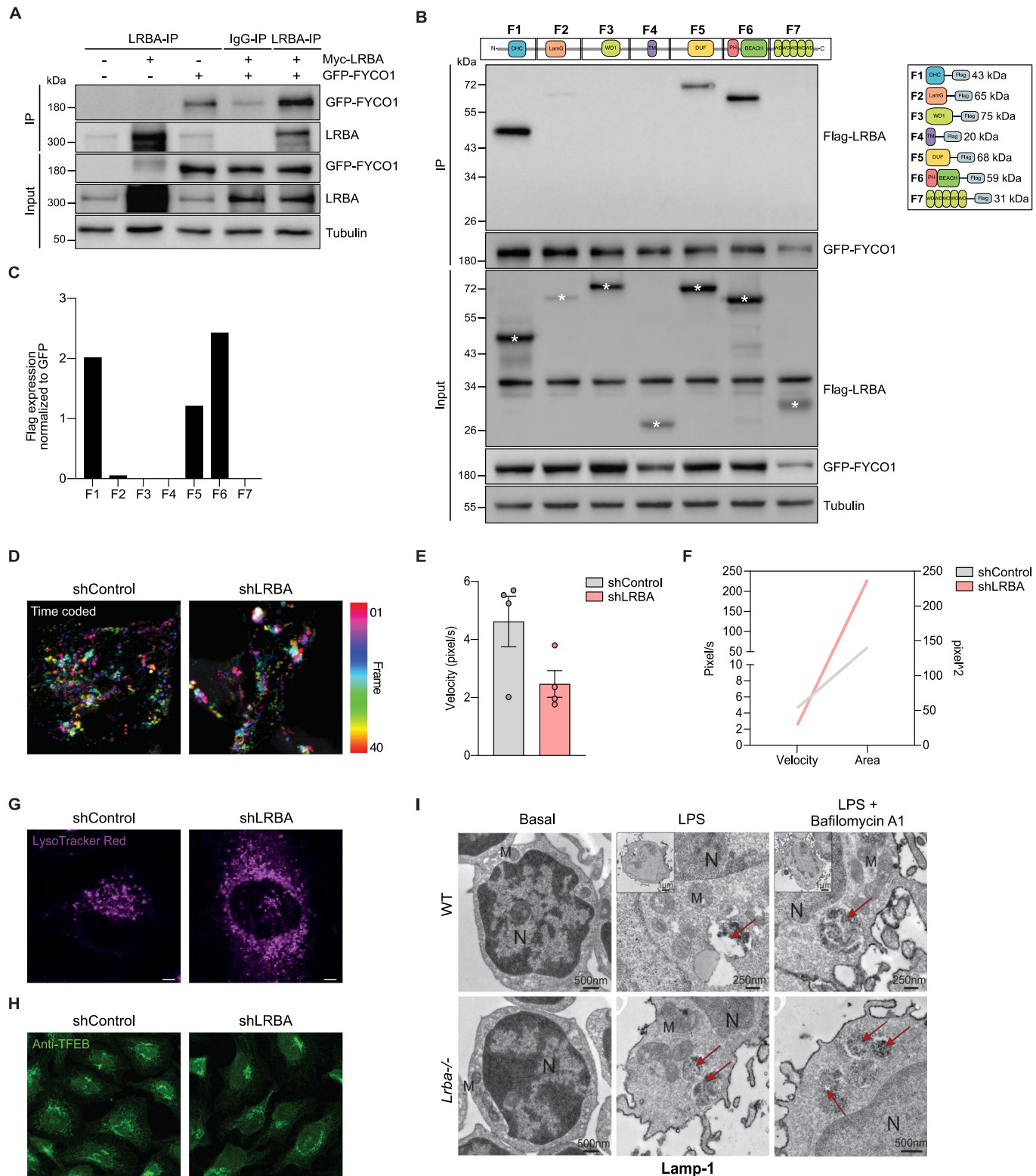

lysosomes via Lamp-1 immunogold labelling revealed an accumulation of lysosomes in LPS-stimulated B cells from $Lrba^{-/-}$ mice compared to WT (Fig. 5I).

Our data indicate that LRBA interacts with the PI3K-III complex and FYCO1, coordinating autophagosome maturation and transport, and supporting efficient autophagic flux.

Figure 5.  LRBA binds to FYCO1 facilitating proper autophagosome movement and lysosome positioning.

(A) WT HEK293T cells were transfected with GFP-FYCO1 and/or Myc-LRBA plasmid. Pull down was performed with anti-LRBA and immunoblotted with anti-GFP. (B) WT HEK293T cells were co-transfected with GFP-FYCO1 plasmid and flag-tagged fragments 1 to 7 of LRBA protein domains. Flag-LRBA pull downs were immunoblotted with anti-GFP. Schematic representation of Flag-tagged plasmids containing different LRBA protein domains is shown on the right. (C) Ratio of Flag expression normalized to GFP expression of $n = 2$ independent biological replicates. (D) Mobility of autophagosomes was evaluated in LC3-GFP transfected shControl and shLRBA HeLa cells by time-lapse movies (Movies EV1 and EV2) acquired from individual frames, color-coded and projected into one image, using the temporal color-coding module of the Zeiss Zen Black software. (E) Quantification of autophagosomes velocity was performed using Manual Track under resting conditions. Each dot corresponds to one autophagosome tracked over 25 frames for shControl (grey) and shLRBA (red) HeLa cells transfected with LC3-GFP. Bars represent mean ± SD from $n = 4$ tracked autophagosome from $n = 1$ videos. (F) Correlation graph of velocity and area of the autophagosomes tracked in shControl (grey) and shLRBA (red) HeLa cells. (G) Lysosomes from shControl and shLRBA HeLa cells under resting conditions were stained with LysoTracker red. Scale bar = 5 µm. (H) Representative fluorescent microscopy images of shControl and shLRBA cells stained with anti-TFEB upon resting conditions. Scale bar = 5 µm. (I) EM images of LPS-stimulated B cells from WT and $Lrba^{-/-}$ mice using immunoperoxidase labeling for Lamp-1. Arrows point to lysosomes. N nucleus, M mitochondria. Images represent higher magnification of the inset's black-boxed area. A total of three mice per genotype were analyzed. Scale bar = 500 nm for WT Basal and $Lrba^{-/-}$ Basal, and $Lrba^{-/-}$ LPS and LPS+Bafilomycin A1. 250 nm for WT LPS and LPS+Bafilomycin A1. Source data are available online for this figure.

## LRBA deficiency enhances MHC class II restricted antigen presentation and T-cell responses

Autophagy has long been associated with immune recognition and responsiveness (Crotzer and Blum, 2009). In particular, it has been identified as a route by which cytoplasmic and nuclear antigens are degraded and loaded onto MHC-II molecules (e.g. HLA-DR) via fusion of autophagosomes with MIIC vesicles in professional and non-professional antigen presenting cells (Arbogast et al, 2019; Ljunggren and Anderson, 1998; Munz, 2016; Runsala et al, 2023). The resulting peptide-MHC-II complexes are then transported to the cell surface and presented to CD4+ T cells, triggering cytokine production and the activation of an adaptive immune response (Munz, 2021).

Given our data that LRBA is essential for autolysosome formation, we considered that LRBA deficiency would impact antigen presentation mediated by autophagy. To test this, we evaluated MHC-II-mediated antigen presentation in vitro. It was previously shown that an influenza matrix protein 1-LC3 (MP1-LC3) fusion protein is targeted to autophagic membranes, enhancing its degradation in MIICs and its MHC-II presentation to CD4+ T cell clones (Schmid et al, 2007). We leveraged this assay by expressing either GFP-LC3 or MP1-LC3 in WT and LRBA-KO HaCat cells (target cell), pre-stimulated them with IFN-γ, and then co-cultured them with or without MP1-specific CD4+ T cells from healthy donors (effector cells). To monitor productive antigen presentation on HaCat cells, we assessed the concentrations of IFN-γ, IL-6, TNF-α and MIP-1β secreted from T cells into the supernatant after 24 h of co-culture (Fig. 6A).

We observed elevated cytokine levels after co-culture of MP1-LC3 WT HaCat cells with CD4+ T cells, which was absent when cultured alone or with GFP-LC3 WT HaCat cells (Figs. 6B,C and EV6A,B). Interestingly, we observed higher cytokine concentrations when T cells were co-cultured with MP1-LC3 LRBA-KO HaCat cells compared to MP1-LC3 WT HaCat cells (Figs. 6B,C and EV6A,B). These data suggest that MHC-II antigen presentation is potentially elevated in HaCat cells lacking LRBA.

To corroborate these findings, we detected the MIIC markers LC3 and HLA-DR in stimulated HaCat cells by fluorescence microscopy. Compared to WT HaCat cells, LRBA-KO HaCat cells displayed a higher number of MIIC vesicles (Fig. 6D,E), and an increased co-localization of HLA-DR and LC3 (Fig. 6D–F). In addition, HLA-DR-positive vesicles were significantly larger in

LRBA-KO HaCat cells compared to WT cells (Fig. 6G). Similar results were obtained when using the MIIC marker, HLA-DM (Fig. EV6C–E). Reconstitution of LRBA-KO HaCat cells with Myc-LRBA restored the number and size of MIIC vesicles, as well as co-localization of HLA-DR and LC3 markers, to WT levels (Figs. 6E–G and EV6F,G). Of note, untreated HaCat cells revealed increased co-localization of HLA-DR and LC3 in LRBA-KO HaCat cells already at basal levels, which was not modified after VPS34 inhibition (Fig. EV6H).

Flow cytometry analysis confirmed the elevated surface presentation of HLA-DR molecules in IFN-γ-stimulated HaCat cells lacking LRBA. This elevation was restored to WT levels upon ectopic expression of Myc-LRBA (Fig. 6H,I).

These data suggest that LRBA deficiency leads to an accumulation of enlarged antigen-containing MIIC vesicles, resulting in enhanced MHC-II-restricted antigen presentation of autophagy substrates to T cells. This, in turn, results in increased production of proinflammatory cytokines driven by T-cell activation.

## Discussion

In this study, we show that LRBA interacts with PIK3R4 to promote PIK3-III activity and autophagic flux. Thus, LRBA deficiency impairs autophagy and leads to an accumulation of enlarged autophagosomes. Similarly, we found that LRBA deficiency leads to an accumulation of MIIC vesicles, which was associated with increased MHC-II restricted antigen presentation and proinflammatory T-cell mediated responses in vitro. We propose that these functions of LRBA contribute to the hyperinflammation seen in LRBA-deficient patients, distinct from aberrant CTLA-4 trafficking. Together, these mechanisms would be expected to increase the severity of LRBA deficiency when compared to CTLA-4 insufficiency (Lo et al, 2016; Taghizade et al, 2023). However, future analyses will be required to disentangle the contribution of each mechanism to the hyperinflammatory disease phenotype arising from LRBA deficiency.

Our data suggest that the LRBA-PIK3R4 interaction occurs via WD40-WD40 domain binding (Jain and Pandey, 2018). WD40 domains are common protein-protein interaction domains, and the WD40 domain in WDFY3/ALFY enable its interaction with PI3K-III complexes and, in turn it participates in cargo selection and delivery to autophagosomes (Cullinane et al, 2013; Isakson et al, 2013). Only

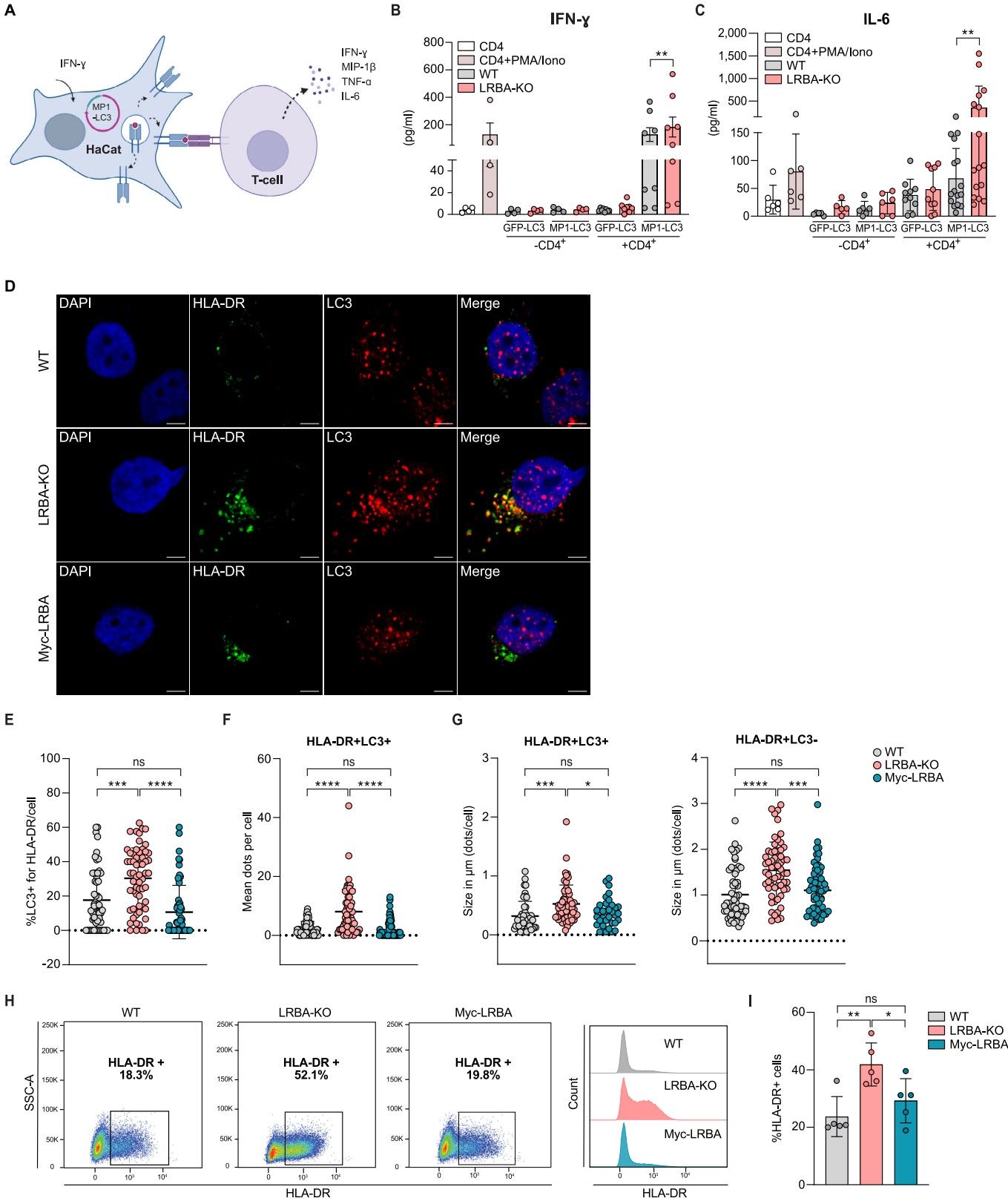

Figure 6.  LRBA deficiency enhances MHC class II restricted antigen presentation and T cell responses.

(A) Schematic illustration of the antigen presentation assay. WT or LRBA-KO HaCat cells stably expressing LC3-GFP or MP1-LC3 (target cells), and pre-treated with IFN-γ for 24 h to up-regulate MHC-II molecules were cultured with a MP1-specific CD4+ T cell clone (effector cells) from a Healthy Donor for 20 h. Following incubation, proinflammatory cytokines were measured in the culture supernatants. (B, C) ELISA assays on culture supernatants from co-cultures of an T-cell effector to HaCat target cell (E:T = 10:5) ratios as described in (A). Each dot represents two technical replicates while bars represent the mean ± SD of (B) IFN-γ (n = 4 individual biological replicates) and (C) IL-6 (n = 4 individual biological replicates) secretion in WT (grey) and LRBA-KO (red) HaCat cells. (D) Representative confocal microscopy images of WT, LRBA-KO and Myc-LRBA reconstituted HaCat cells stimulated overnight with IFN-γ and treated with 20 μM chloroquine for 6 h. Fixed cells were stained for HLA-DR (green), LC3 (red) and DAPI (blue) for the nuclei. Scale bar = 5 μm. (E–G) Scatter plots represent the (E) percentage of LC3+ dots also positive for HLA-DR per cell, (F) the number of dots positive for HLA-DR and LC3 (known as MIIC vesicles), and (G) the average size in μm of HLA-DR+/LC3+ and HLA-DR+/LC3- in WT (grey) LRBA-KO (red) and Myc-LRBA (teal) HaCat cells. Each dot represents one cell from n = 3 individual biological replicates with a total of at least 30 cells analyzed per condition, mean ± SD shown by the black line and error bars. Quantification of LC3, HLA-DR or double positive dots per cell was performed using a semiautomatic plugin designed on FIJI. (H, I) HLA-DR expression in HaCat cells. (H) Representative dot plots of HLA-DR expression in WT, LRBA-KO and Myc-LRBA HaCat cells. (I) WT (grey), LRBA-KO (red) and Myc-LRBA (teal) HaCat cells positive for HLA-DR. Dots represent the mean of n = 2 technical replicates while bars represent the mean ± SD of n = 5 independent biological replicates. Statistical analysis of (B, C) was performed using a ratio paired student t-test and for (E–G, I) a one-way ANOVA with Tukey's multiple comparisons test, *P < 0.05 ((G) HLA-DR+LC3+: P = 0.0483), ((I): P = 0.0471), **P < 0.01 ((B): P = 0.0022), ((C): P = 0.0013), ((I): P = 0.0058), ***P < 0.001 ((E): P = 0.0002), ((G): HLA-DR+LC3+: P = 0.0008), ((G): HLA-DR+LC3−: P = 0.0004), ****P < 0.0001. Source data are available online for this figure.

PIK3R4 harbours a WD40 domain within the PI3K protein family (Bilanges et al, 2019). Our data also suggest that the BEACH domain of LRBA, which is involved in the trafficking of CTLA-4-containing vesicles in Tregs (Lo et al, 2015), interacts with FYCO1. These data suggest that LRBA forms distinct protein complexes through its BEACH and WD40 domains, playing a role in both CTLA-4 recycling (Lo et al, 2015) and autophagy. It will be of interest to determine whether LRBA protein-protein interactions are cell- or stimuli-specific.

LRBA-KO cells showed reduced PI(3)P levels after autophagy induction. However, both WT and LRBA-KO cells produced similar levels of PI(3)P in resting cells, suggesting that the PI(3)P constitutive pool is not affected by the loss of the LRBA-PIK3R4 interaction. Indeed, downregulation of VPS34 expression in resting glioblastoma cells resulted in a residual production of PI(3)P (Johnson et al, 2006), indicating that the PI(3)P constitutive pool is generated independently of the PI3K-III complex. (Falasca and Maffucci, 2009; Gaullier et al, 2000; Johnson et al, 2006). VPS34-KO cells showed normal vesicle trafficking between the trans-Golgi and late endosomes, normal endocytic uptake of fluid-phase markers, and normal association with early endosomes, but disrupted late endosomal trafficking (Sarbassov et al, 2004). Therefore, the effect of LRBA deficiency in CTLA-4 vesicle trafficking could also be related to an aberrant activity of the PI3K-III complex. Moreover, we demonstrated that LRBA interacts with UVRAG, which together with PIK3R4 and VPS34 forms a protein complex located at the membrane of early endosomes acting in endocytosis, cytokinesis and lysosome recycling, suggesting that LRBA might also affect other vesicle trafficking processes. PI(3)P acts as a membrane-bound localized signal, controlling the assembly of PI(3)P-binding scaffold proteins such as WIPI2 and DFCP1 that mediate autophagosome biogenesis (Marat and Haucke, 2016). WIPI2 is necessary for recruiting the lipidation machinery for the phagophore-forming protein LC3B (Dooley et al, 2014; Polson et al, 2010), whereas the ATPase DFCP1 is necessary for releasing the newly formed autophagosomes from the ER (Nahse et al, 2023). LRBA-KO cells and WT cells displayed comparable levels of LC3 lipidation, despite reduced WIPI2 and DFCP1 puncta in LRBA-KO cells ruing autophagy. Normal autophagosome formation was also seen in DFCP1-mutant cells and in Vps34-mutant cells, despite reduced WIPI2 puncta (Bilanges et al, 2017). Similar to our observations, shVPS34 glioblastoma cells as well as vps34-KO MEF cells showed enlarged late endosomes but with an intact capacity to fuse with lysosomes (Johnson et al, 2006). Our data, however, suggest a delay of autophagosome/lysosome fusion in LRBA-KO cells, which highlights the additional role played by LRBA in the autophagy process.

We suggest that the LRBA-FYCO1 interaction is essential for autophagosome mobilization and regular lysosome positioning followed by autophagosome/lysosome fusion. This notion is reinforced by previous reports showing the importance of FYCO1 for intracellular transport of autophagic vesicles (Chen et al, 2011; Pankiv et al, 2010; Pankiv and Johansen, 2010), lysosomes and phagosomes (Mrakovic et al, 2012; Pu et al, 2016). In fact, FYCO1 depletion led to an accumulation of autophagosomes as observed in LRBA-KO cells (Behrends et al, 2010). Since ectopic expression of Myc-LRBA in LRBA-KO HEK293T cells rescued the recruitment of DFCP1 and the degradation of p62, our data support the requirement of LRBA for autophagosome maturation and therefore autophagy flux. Intriguingly, LRBA plays a role in ATG9A vesicle trafficking to mitochondria in HeLa cells and loss of LRBA resulted in reduced mitophagy efficiency (Nguyen et al, 2021).

We previously reported defective autophagy in naive B cells from LRBA-deficient patients, which was associated with poor cell survival, reduced plasmablast differentiation and low antibody production (Lopez-Herrera et al, 2012). The importance of autophagy for plasmablast survival and differentiation has been demonstrated in Atg5-deficient mice (Pengo and Cenci, 2013; Pengo et al, 2013). In T lymphocytes, the PI3K-I complex plays a key role in autophagy upon starvation and T cell receptor stimulation, whereas PI3K-III complex activity is dispensable (McLeod et al, 2021). However, PI3K-III complex-dependent autophagy is required for naive T-cell home-ostasis through the quality control of mitochondria (Willinger and Flavell, 2012). Together, these observations may explain the impaired activation of mTORC1 and mTORC2 complexes in human LRBA-deficient Tregs (via PI3K-I complex) and the skewed T-cell memory phenotype (via PI3K-III complex) in LRBA deficiency (Charbonnier et al, 2015). LRBA-deficient patients have normal numbers of circulating T cells, suggesting that abnormal autophagy does not affect T-cell survival, but it nevertheless might affect endocytic trafficking routes.

Previous findings showed that the LC3-lipid and Atg12-Atg5 systems are required in HaCat cells for targeting MP1-LC3 fusion proteins for MHC class II loading and enhanced MHC class II presentation to CD4+ T-cells (Schmid et al, 2007). Using this setup in LRBA deficiency, HaCat cells showed enlarged and augmented number of MIIC vesicles potentially associated to an

autophagosome rerouting towards MIIC vesicles as a cause of an autophagosome-lysosome fusion defect. Similar autophagosome enlargement and accumulation of the autophagy cargo protein p62 was also observed in LRBA-KO HEK293T and shLRBA HeLa cells. Ectopic expression of WT-LRBA in LRBA-KO cells restored MIIC vesicle sizes and number to those observed in WT cells.

MP1-specific CD4$^+$ T-cell clones co-cultured with LRBA-KO HaCat cells displayed increased proinflammatory cytokine secretion, indicating enhanced CD4$^+$ T-cell activation. These observations are in line with the clinical picture of LRBA-deficient patients, which is characterized by T-cell immune dysregulation (Alkhairy et al, 2016; Azizi et al, 2017a; Gamez-Diaz et al, 2016). An enhanced proinflammatory response in the context of LRBA deficiency has also been reported for $Lrba^{-/-}$ mice, which display increased susceptibility to DSS-induced colitis (Wang et al, 2019). Dendritic cells from $Lrba^{-/-}$ mice exhibited excessive IRF3/7- and PI3K/mTORC1-dependent signaling and type I IFN production in response to the stimulation of the Toll-like receptors (TLRs) 3, TLR7, and TLR9. It would however be valuable to determine whether FYCO1 dysfunction alone leads to enhanced cargo in MIIC vesicles and antigen presentation, which could clarify whether the observed MIIC accumulation is primarily dependent on FYCO1 or if it is solely due to LRBA loss.

Finally, we show that LRBA-deficient cells exhibit concomitant high mTORC1 activity potentially contributing to defective autophagy (Egan et al, 2011). Thus, mTORC1 inhibitors may be valuable treatment options for LRBA-deficient patients (Brachmann et al, 2009; Goodman et al, 2014). In fact, several clinical reports on patients with LRBA deficiency demonstrated that treatment with Sirolimus, an mTOR inhibitor, improved their clinical symptoms (Azizi et al, 2017b; Gamez-Diaz et al, 2016; Seidel et al, 2018).

In conclusion, our data suggest that LRBA facilitates autophagy by interacting with PIK3R4 and FYCO1. Loss of LRBA resulted in low PI(3)P production, reduced autophagosome-lysosome fusion, enlarged autophagosomes, and inefficient degradation of cargo proteins via autophagy. Accumulation of cargo/antigenic peptides, particularly in MIIC vesicles that contain MHC-II molecules, was associated with enhanced antigen presentation in LRBA-deficient cells, leading to a stronger T-cell driven proinflammatory immune response.

# Methods

### Reagents and tools table

| Reagent/resource | Reference or source | Identifier or catalog number |
|---|---|---|
| **Experimental models** | | |
| HEK 293T cell (*H. sapiens*) | ATCC | CRL-11268 |
| HaCat cells (*H. sapiens*) | Kind gift from Prof. Rajiv Khanna, Brisbane, Australia. | Generated in the laboratory of Prof. Norbert E. Fusenig (J. Cell Biol. 1998) |
| HeLa cells (*H. sapiens*) | ATCC | CCL-2 |
| LCL (Lymphoblastoid cell line (*H. sapiens*) | Generated in our laboratory using Healthy donor cells and Epstein Barr Virus supernatant | N/A |

| Reagent/resource | Reference or source | Identifier or catalog number |
|---|---|---|
| MP1-specific CD4+ T cell clone | Generated in the laboratory of Prof. Christian Münz, University of Zürich, Switzerland | |
| Ramos B cells | ATCC | CRL-1596 |
| C57BL/6 N *Lrba* −/− | Generated in the laboratory of Prof. Manfred Kilimann. Max-Planck Institute Göttingen, Germany | C57BL/6N-Lrba^tm1.1Kili/leg RRID:IMSR_EM:14924 |
| Naive B cells (*M. musculus*) | From C57BL/6 N WT and Lrba−/− mice | N/A |
| **Recombinant DNA** | | |
| pmCherry-GFP-LC3 | Dr. Ian Gentle University of Freiburg, Germany | N/A |
| pClneo FLAG | Promega | E1841 |
| Myc-DDK LRBA | Origene | RC17204 |
| pVitro-hygro-N-myc-hVps34/Vps15-C-V5-his | Addgene | 24296 |
| pEGFP-Atg14L | Addgene | 21635 |
| pEGFP-C1-hUVRAG | Addgene | 24296 |
| pClneo FLAG LRBA fragments | Dr. Bernice Lo National Institute of Health, USA | N/A |
| pGFP-FYCO1 | Dr. Christian Behrends University of Frankfurt, Germany | N/A |
| GFP-LC3 | Prof. Christian Münz, University of Zürich, Switzerland | N/A |
| MP1-LC3 | Prof. Christian Münz, University of Zürich, Switzerland | N/A |
| **Antibodies** | | |
| Rabbit anti-LC3 | Cell Signaling | E5Q2K |
| Mouse anti-Lamp1 | Invitrogen | MA1-205 |
| Mouse anti-WIPI2 | Abcam | ab105459 |
| Rabbit ZFYVE/DFCP1 | LS-bio | LS-B15646-50 |
| Rabbit anti-GAPDH | Cell Signalling | 2118 |
| Mouse anti-FLAG | Sigma-Aldrich | SAB4301135 |
| Mouse anti-GFP | Santa Cruz | SC-9996 |
| Mouse anti-AMPK total | Cell Signaling | 2793S |
| Rabbit anti-pAMPK (T172) | Cell Signaling | 2535S |
| Rabbit anti-LC3 | Novus | NB110-2220 |
| Rabbit anti-LRBA | Sigma-Aldrich | HPA023597 |
| Rabbit anti-Myc | Cell Signaling | 2278S |
| Mouse anti-PIK3R4 | Novus | H00030849-M02 |
| Rabbit anti-p62 | Enzo Lifescience | 5114S |
| Mouse anti-p62 AF647 | Abcam | ab194721 |

| Reagent/resource | Reference or source | Identifier or catalog number |
|---|---|---|
| Rabbit anti-p70S6K total | Cell Signaling | 2708S |
| Mouse anti-p-p70S6K (T389) | Cell Signaling | 9206S |
| Mouse anti-HLA-DR | Cell signaling | NB-100-7785 |
| Mouse anti-HLA-DM AF488 | Santa Cruz | sc-32248 AF488 |
| Rabbit anti-FYCO1 | Santa Cruz | 25730-1-AP |
| Rabbit anti-P6RP | Cell signaling | 4858 |
| Rabbit anti-VPS34 | Novus | NB110-87320SS |
| Mouse anti-HLA-DR FITC | Invitrogen | A16118 |
| Rabbit anti-pS6RP Pacific Blue | Cell signaling | 8520S |
| Anti-rabbit IgG-FITC | Invitrogen | A16118 |
| Anti-rabbit AF488 | Cell Signaling | 4412 |
| Anti-rabbit AF555 | Cell Signaling | 4413 |
| Anti-mouse AF647 | Cell Signaling | 4410 |
| HRP-linked anti-mouse | Santa Cruz | sc-516102 |
| HRP-linked anti-rabbit | Invitrogen | 31470 |
| HRP-linked anti-Tubulin | Prointech | HRP-66031 |
| PLA probe anti-mouse minus | Sigma-Aldrich | DU92004 |
| PLA probe anti-goat minus | Sigma-Aldrich | DU92006 |
| PLA probe anti-rabbit plus | Sigma-Aldrich | DUO92101 |
| **Oligonucleotides and other sequence-based reagents** | | |
| gRNA for CrispR | IDT | N.A. |
| shControl | Thermo Fisher | |
| shLRBA | Thermo Fisher | |
| **Chemicals, enzymes and other reagents** | | |
| 3'3-diaminobenzidine | Sigma-Aldrich | D12384 |
| ABC Elite Kit | BIOZOL | VEC-PK-6100 |
| Bafilomycin A1 | InvivoGen | tlrl-baf1 |
| B cell isolation kit, mouse | Miltenyi Biotec | 130-090-862 |
| BD Cytofix/Cytoperm solutionTM | BD | 554714 |
| BSA Albumin Bovine V | Serva | 11903.03 |
| Chloroquine | Sigma-Aldrich | C6628 |
| DAPI | Sigma-Aldrich | D9542 |
| DMEM high glucose | Life Technologies GmbH | 31966047 |
| DPBS | Sigma-Aldrich | D8537 |
| Duolink In Situ Mounting Medium with DAPI | Merck | DEU82040 |
| Duolink PLA Kit | Sigma-Aldrich | DUO92008 |
| Durcupan resin | Sigma-Aldrich | 44611 |
| Dynabeads protein G | Thermo Fisher Scientific | 10004D |

| Reagent/resource | Reference or source | Identifier or catalog number |
|---|---|---|
| EBSS | Thermo Fisher | 14155063 |
| EDTA-free protease inhibitor cocktail | Roche | 5,893E + 09 |
| Ethanol | VWR | 20.821.310 |
| Ethylenediaminetetraacetic acid solution (EDTA) | Sigma-Aldrich | E8008 |
| Fetal bovine serum (FBS) | Thermo Fisher | F7524 |
| Glutraldehyde | Sigma-Aldrich | 354400 |
| Goat serum | Sigma-Aldrich | G9023 |
| HEPES | Sigma-Aldrich | H0887 |
| Hydrochloric acid (HCl) | AppliChem | 182883.12 |
| IFN-γ ELISA Kit | MABTECH | 3420-1H-6 |
| IL-6 ELISA Kit | MABTECH | 3460-1HP-1 |
| Lead(II) acetate trihydrate | Merck Millipore | 107375 |
| Lipofectamine™ 2000 | Invitrogen | 11668-027 |
| LPS from *E. coli* O55 | Sigma-Aldrich | L2654 |
| Lysotracker Red | Invitrogen | L12492 |
| Methanol | Honeywell | 32213 |
| MG132 | InvivoGen | tlrl-mg132-2 |
| Milk powder | Roth | T145.1/.3 |
| MIP-1β ELISA Kit | Thermo Fisher | 88-7034-22 |
| NP-40 | Calbiochem | 492018 |
| Opti-MEM™ - Reduced Serum Medium | Thermo Fisher | 31985070 |
| Optiprep | Sigma-Aldrich | D1556 |
| Osimum tetroxide (OsO4) | Sigma-Aldrich | 201030 |
| Paraformaldehyde | Sigma-Aldrich | 158127 |
| Penicillin-Streptomycin (10 000U/ml) | Thermo Fisher | 15140122 |
| Perm/Wash BufferTM | BD | 554723 |
| PI(3)P Mass ELISA kit | Echelon | K-3300 |
| Pierce™ BCA Protein Assay Kit | Thermo Fisher | 23225 |
| Potassium hydroxide (KOH) | Sigma-Aldrich | 6103 |
| Propylene oxide | Sigma-Aldrich | 75-56-9 |
| Proteinase K | Zymo Research | D3001-2 |
| PVDF membrane | Bio-Rad Laboratories GmbH | 1620177 |
| Rapamycin | InvivoGen | tlrl-rap |
| Recombinant human IFN-γ | Peprotech | 300-02 |
| RPMI 1640 + L-Glutamine | Sigma-Aldrich | 21875091 |
| Saponin | Sigma-Aldrich | 47036 |
| Signal Fire/Plus chemiluminescent substrates | Cell Signaling | 12630S |

| Reagent/resource | Reference or source | Identifier or catalog number |
|---|---|---|
| Silver enhancement HQ-Silver kit | Nanoprobe | 2012 |
| Sodium chloride (NaCl) | Roth | 9265.1 |
| Sodium deoxycholate | Sigma-Aldrich | 89904 |
| Sodium dodecyl sulfate (SDS) | Sigma-Aldrich | 436143 |
| Sodium pyruvate | Sigma-Aldrich | P5280 |
| Sucrose | Sigma-Aldrich | 84097 |
| Sulfuric acid ($H_2SO_4$) | Sigma-Aldrich | 339741 |
| TMB substrate | MABTECH | 3652-F10 |
| TNF-α ELISA Kit | MABTECH | 3512-1HP-1 |
| Torin1 | InVivoGen | inh-tor1 |
| Trichloroacetic acid (TCA) | Sigma-Aldrich | 8.223.420.250 |
| Tris | Applichem GmbH | A1379 |
| Triton X-100 | Sigma-Aldrich | 93443 |
| Trypsin-EDTA (0.05%), phenol red | Thermo Fischer | 25300-054 |
| Tween 20 | Sigma-Aldrich | P9416 |
| Uranyl acetate | Polysciences | 21447-25 |
| Viability Dye eFluor780 | Invitrogen | 65-0865-14 |
| VPS34-In1 | Selleckchem | S7980 |
| Wortmannin | InVivogen | tlrl-wtm |
| **Software** | | |
| Affinity designer 2 | Serif | N/A |
| Duolink Image Tool | Sigma-Aldrich | |
| FlowJo | Treestar Inc | |
| FUNCBase database | | |
| GraphPad Prism 10 | | |
| High Ambiguity Driven Biomolecular DOCKing, version 2.2 | | |
| Homology modelling software Prime | Schrödinger Suite 2018-1, LLC | |
| ImageJ/FiJi | | |
| ImageSP | | |
| Imaris9.7 | Oxford Instrument | |
| STRING database | | |
| Zeiss blue/black | Zeiss | |
| **Other** | | |
| BD FACS canto II | Beckton Dickinson | |
| Dounce homogenizer | WHEATON® Dounce Tissue Grinder | |
| Fusion SL device | Peqlab | |
| SP8 | Leica | |
| Zeiss LSM 710 | Zeiss | |
| Zeiss LSM 880 | Zeiss | |
| Zeiss LEO 906 transmission electron microscope | Zeiss | |

## Study protocol

Collection of peripheral blood mononuclear cells from healthy donors (HD) and LRBA-deficient patients to generate LCL cell lines was approved by the ethics committee of the University of Freiburg, Germany, vote n° 290/13. Patients have signed an informed consent, and experiments were performed conformed to the principles set out in the WMA Declaration of Helsinki and the Department of Health and Human Services Belmont report.

## Generation of a LRBA-KO HEK293T cell line, a HaCat and a Ramos cell line by CRISPR-Cas9, and a LRBA knock-down HeLa cell line by shRNA (shLRBA cells)

LRBA-KO HEK293T, HaCat and Ramos cells were generated using the Alt-R-CRISPR-Cas9 system according to the manufacturer´s instructions (www.idtdna.com). The gRNA CCACCAACAGGT-GATGACGG specific for exon 2 of human LRBA, was inserted into the cells by electroporation together with the crRNA: tracrRNA complex and the Cas9. Following 48 h incubation, cells were single cell sorted by flow cytometry. LRBA-KO clones were validated by western blot for LRBA expression abolishment and by Sanger sequencing for mutation identification (Fig. EV7A–F). shLRBA HeLa cells were generated by lentiviral transfection with the vector pLKO.1 (SIGMA) containing the shRNA sequence: CCGGGCAGAAGATATTCACAGACATCTCGAGATGTCTGT-GAATATCTT-CTGCTTTTTTG (TRCN0000148136, SIGMA) according to the manufacturer's instructions (Fig. EV7G). Ramos cells were authenticated by STR profiling and all cells lines were tested weekly for Mycoplasma contamination.

## Human lymphoblastoid B cell line generation

LCL cells were generated from B cells from three HD and two LRBA-deficient patients (P1: c.2004+2 A > G; P2: p.S2713Hfs) after isolation from PBMCs by negative selection, and incubation 1:1 cell: EBV containing-media for four days. LCL actively proliferate, secrete antibodies and express LRBA (Fig. EV7H).

## Mice

*Lrba* knock-out mice (*Lrba*$^{-/-}$) were kindly provided by Prof. Manfred Kilimann, Max-Planck-Institut für Experimentelle Medizin (MPIEM), Göttingen, Germany. They were generated on a C57BL/6N background by homologous recombination producing a loss-of-protein deletion of exon 4 of *Lrba* (C57BL/6N-Lrbatm1.1-Kili/Ieg RRID:IMSR_EM:14924) (Kurtenbach et al, 2017). Mice were bred and maintained on a C57BL/6N background under specific pathogen-free (SPF) conditions at the animal facility of the Center for Experimental Models and Transgenic Service (CEMT), University Medical Center Freiburg, Germany. All animal experiments were approved by the local animal ethics committee (Regierungspräsidium Freiburg, Germany) under the following reference numbers: X-15/05F, G-16/19, G-16/94 and G15-168.

## Plasmids

Human LRBA was cloned in seven different fragments into pCIneo FLAG vectors (Promega). These plasmids were kindly provided by

Dr. Bernice Lo from the National Institute of Health, Bethesda, USA (Lo et al, 2015). mCherry-GFP-LC3 plasmid was a kind gift from Dr. Ian Gentle from the University of Freiburg, Germany, whereas GFP-FYCO1 was kindly provided by Dr. Christian Behrends from the University of Frankfurt, Germany. Human LRBA full-length tagged to Myc-DDK was purchased in Origene. pVitro-hygro-N-myc-hVps34/Vps15-C-V5-his-plasmid (Addgene), pEGFP-Atg14L (Addgene) and pEGFP-C1-hUVRAG (Addgene) were provided by Jonathan Backer (Yan et al, 2009), Tamotsu Yoshimiri (Matsunaga et al, 2009) and Noburo Mizushima (Itakura et al, 2008), respectively. GFP-LC3 and MP1-LC3 plasmids were previously generated in the laboratory of Prof. Christian Münz from the University of Zürich, Switzerland.

## Reconstitution of LRBA

Rescue experiments were conducted in LRBA-KO HEK293T and HaCat cell lines transfected with 2 µg of WT LRBA plasmid tagged with Myc (Origen) using Lipofectamine 2000 (Invitrogen). These reconstituted cells lines are called Myc-LRBA in this manuscript. Details of the transfection protocol are described in the "Transfections and Immunoprecipitations" section below.

## Proximity ligation assay

In situ PLA was performed using Duolink kit (Sigma-Aldrich) in LCL cells from HD 1 and 2, and Patient 1 and 2. Cell fixation and permeabilization was performed with 4% PFA and 0.1% Triton X-100, respectively. Incubation with primary antibodies was followed by incubation with secondary antibodies that are conjugated with oligonucleotides, PLA probe anti-mouse (or anti-goat) MINUS and PLA probe anti-rabbit PLUS (Sigma-Aldrich). After ligation with DNA oligonucleotides and amplification with a DNA polymerase, the amplified product was detected as a fluorescent signal with a confocal microscope (Zeiss LSM700). Signal quantification was performed using Duolink Image Tool (Sigma-Aldrich).

## Transfections

HEK293T cells were co-transfected with 2 µg of one of the seven Flag-tagged-LRBA-fragment plasmids or Myc-LRBA plasmid, plus 2 µg of Myc-PIK3R4/His-VPS34, GFP-FYCO1, GFP-ATG14L or GFP-UVRAG using Lipofectamine 2000 (Invitrogen) according to the manufacturer's protocol. Cells were harvested after 48 h and lysed with IP buffer (50 mM Tris-HCl pH 6.8, 150 mM NaCl, 0.2% NP-40, 1 mM EDTA), plus 1× complete EDTA-free protease inhibitor cocktail (Roche). Lysates were collected and used for immunoprecipitation experiments.

## Immunoprecipitations

LRBA IP, or flag-tagged LRBA IP or GFP IP were performed using 200 µg of cell lysates from HEK293T cells and 1 µg of either rabbit anti-LRBA (Sigma-Aldrich), or mouse anti-FLAG antibody (Sigma-Aldrich), mouse anti-PIK3R4 (Novus) or mouse anti-GFP (Santa Cruz). Then, 40 µl of Dynabeads protein G (Thermo Fischer Scientific) were added to the lysate/antibody mix and incubated overnight. Beads were washed with lysis buffer, and proteins were eluted with 2% SDS and resuspended in Laemmli buffer. In total, 20 µl of the eluted proteins were separated by SDS-PAGE, blotted and detected by immunoblotting.

## Immunoblotting and antibodies

Cell lysates were generated with RIPA buffer (50 mM Tris, 1% NP-40, 0.5% sodium deoxycholate, 100 mM NaCl, 1 mM EDTA, 0.1% SDS) + 1× complete EDTA-free protease inhibitor cocktail (Roche). Total protein concentrations were determined by bicinchoninic acid Protein Assay (Thermo Fisher Scientific). Protein lysates were size-fractionated by SDS-PAGE and electro transferred to a PVDF membrane in a wet blotting system for 1.5 h at 45 V. After blocking with 5% milk in TBST (20 mM Tris, 150 mM NaCl, 0.1% Tween 20), the membranes were incubated at 4 °C with any of the following primary antibodies: anti-AMPK total (Cell Signaling), anti-AMPK (pT172; Cell Signaling), anti-DFCP1 (LS-bio), anti-FLAG (Sigma-Aldrich), anti-FYCO1 (Proteintech), anti-GFP (Santa Cruz), anti-LC3 (Novus), anti-LRBA (Sigma-Aldrich), anti-Myc (Cell Signaling), anti-PIK3R4 (Novus), anti-p62 (Cell Signaling), anti-S6K total (Cell Signaling), anti-S6K70 (pT389; Cell Signaling), anti-P6RP (Cell Signalling), anti-VPS34 (Novus-bio), anti-WIPI2 (Abcam). After overnight incubation with the primary antibodies, membranes were immunodetected with their corresponding secondary HRP-coupled antibody (Santa Cruz). Membranes were washed and developed with Signal Fire or Signal Fire Plus chemiluminescent substrates (Cell Signaling). HRP-linked anti-Tubulin (Proteintech) and anti-GAPDH (Cell Signaling) were used as a loading control. Peroxidase activity was detected with the Fusion SL device (Peqlab).

## Autophagy induction

Naive B cells (CD43⁻B220⁺CD3⁻) were obtained by negative selection (mouse CD43 Ly-48 MicroBeads; MACS Miltenyi Biotec) from spleens of WT and $Lrba^{-/-}$ mice, followed by stimulation with LPS from $Escherichia coli$ 055: B5 (20 µg/ml, Sigma-Aldrich) for 3 days in complete RPMI medium (L-glutamine, 10% FCS, 100 µg/mL strepto-mycin, 100 U/mL penicillin, 10 mM HEPES and 1 mM sodium pyruvate) with or without 100 nM of Bafilomycin A1 (InvivoGen) for 3 h. Next, cell pellets were either used for LC3-II detection by immunoblotting or for morphology and LRBA cellular localization analyses by electron microscopy. shControl and shLRBA HeLa cells were seeded at 0.3*10⁶ cells in six-well plates followed by treatment for 16 h or 24 h with 5 µM of MG132 (InvivoGen) alone or in combination with 100 nM of Bafilomycin A1 for 3 h.

## Measurement of phosphatidylinositol-3 phosphate (PI(3)P)

WT and LRBA-KO HEK293T cells were seeded and treated with 333 nM of Torin 1 (Invivogen) or with 1 nM of VPS34-IN (Biomol) for 4 h. After cell collection, cells lysis and extraction of neutral and acidic lipids, PI(3)P was obtained from the organic phase as described before (Chicanne et al, 2012). Finally, PI(3)P was measured using the PI(3)P Mass ELISA kit (Echelon) according to the manufacturer's protocol.

## Fluorescence microscopy

To visualize autophagosomes, shWT and shLRBA HeLa cells were transfected with GFP-LC3 using Lipofectamine 2000 (Invitrogen). Lysosomes were visualized with LysoTracker Red (Invitrogen),

according to the manufacturer's instructions. To visualize autophagosomes and autolysosomes, WT and LRBA-KO HEK293T cells were either stably transduced with the mCherry-GFP-LC3 tandem vector (kindly provided by Dr. Ian Gentle, University of Freiburg), or stained endogenous LC3B. Cells were treated for 4 h with 333 nM Torin 1 (Invivogen) alone or in combination with 100 nM Bafilomycin A1 (Invivogen). To visualize endogenous expression of DFCP1 and WIPI2, WT and LRBA-KO HEK293T cells were treated for 1 h with 50 μM Rapamycin (Invivogen), 333 nM Torin 1 (Invivogen), 100 nM Wortmannin (Invivogen) or incubated in EBSS medium (Thermo Fisher) for starvation. After treatments, cells were fixed for 10 min with 4% PFA (Sigma-Aldrich) and permeabilized with either 0.05% Saponin (Sigma-Aldrich) or 0.1% Triton X-100. Endogenous LC3 expression was detected after 1 h incubation with anti-LC3B (Novus), followed by 45 min incubation with secondary antibodies anti-rabbit-AlexaFluor555 (Cell Signaling). DFCP-1 and WIPI2 expression was detected after overnight incubation with anti-DFCP1 (LS-bio) or anti-WIPI2 (Abcam) antibodies, followed by 1 h incubation with anti-mouse or anti-rabbit Alexa Fluor 488 or Alexa Fluor 647 antibodies (Cell Signaling). After staining, cover slides were mounted with DuoLink In Situ Mounting Medium with DAPI (Merck). The images were acquired using Zeiss LSM710 or LSM880 confocal microscope. The number of dots per cells and co-localization was assessed with ImageJ. The autophagosome size was assessed using Imaris 9.7 (Oxford Instruments). Autophagosomal size was determined from binary images using the analyses particle module of FIJI software (version X), with a particle size from 0 to 20 μm². Autophagosomal mobility was evaluated by time-lapse movies of autophagosomes acquired in individual frames color-coded and projected into one image, using the temporal color-coding module of the Zeiss Zen Black software. Autophagosomes were followed for 25 frames and area and velocity were quantified in pixels.

## LC3B co-localization with HLA-DR molecules in MIICs experiments

HaCat cells were seeded in glass cover slips placed inside 24-well plate and stimulated overnight with IFN-γ to induce MHC-II expression. Cells were treated 6 h with 20 μM of chloroquine, and then with 4% PFA for 15 min at RT in the dark. Permeabilization was performed with 0.1% Triton X-100 for 5 min at RT, followed by three PBS washes. Then cells were saturated with blocking buffer (1% FBS-PBS) for 1 h at RT. Primary antibodies were diluted in the blocking buffer and incubated with the cells for 1 h at RT. Primary antibodies used: rabbit anti-LC3B (Novus), mouse anti-HLA-DR (Novus), mouse anti-HLA-DM AF488 (Santa Cruz) and mouse anti-p62 AF647 (Abcam). Three PBS washes were performed before staining with Alexa Fluor 488/555-conjugated goat anti-mouse or anti-rabbit (Thermo Fischer) for 1 h at RT. After three PBS washes, cell nuclei were stained with DAPI for 5 min prior to mounting the cover slip onto a glass slide using Duolink In Situ Mounting Medium with DAPI (Merck). Cells were visualized through 63×, 1.4 NA oil immersion objective with a confocal laser scanning microscope (LSM710, Zeiss) and images were processed with Fiji software.

## Membrane flotation assay

Cells from three 15-cm dishes were washed twice with PBS and harvested. The cell pellets were collected after centrifugation at

$800 \times g$ for 5 min and resuspended in homogenization buffer (250 mM sucrose, 20 mM HEPES-KOH pH 7.4, 1 mM EDTA and complete EDTA-free protease inhibitor cocktail (Roche). Cells were lysed by 40 strokes in a glass Dounce homogenizer (WHEATON® Dounce Tissue Grinder). Unbroken cells and debris were removed by two centrifugation steps at $2000 \times g$ for 5 min. The supernatant was mixed with an equal volume of 50% OptiPrep (D1556-250ML; Sigma-Aldrich) in homogenization buffer. Discontinuous Optiprep gradients were prepared as described previously (Matsui et al, 2018), in SW41 tubes (344059; Beckman Coulter) by overlaying the following Optiprep solutions in homogenization buffer: 2.4 ml of the diluted sample (25% Optiprep), 1.8 ml of 20% Optiprep, 2 ml of 15% Optiprep, 2 ml of 10% Optiprep, 2 ml of 5% Optiprep and 2 ml of homogenization buffer without Optiprep. The gradients were spin and 13 fractions of 0.95 ml were collected from the top and subjected to TCA precipitation. The final pellet was resuspended in sample buffer and incubated at 95 °C for 5 min.

## Proteinase protection assay

Cells were treated for 2 h with EBSS and 300 nM Torin 1. Afterwards they were washed with PBS and collected by centrifugation at $500 \times g$ for 5 min. Pellets were resuspended in homogenization buffer (250 mM sucrose, 20 mM Hepes-KOH pH 7.4, 1 mM EDTA and complete EDTA-free protease inhibitor cocktail) and lysed by 30 passages with a 25 G needle. After two preclearing steps ($2000 \times g$, 5 min), cell membranes were pelleted by centrifugation at $20,000 \times g$ for 30 min. The pellet was resuspended in 100 μl of homogenization buffer without EDTA or protease inhibitors, divided into three equal fractions and incubated in the presence or absence of proteinase K (100 μg per ml of sample) with or without 0.5% Triton X-100 for 30 min on ice. The samples were then subjected to TCA precipitation and resuspended in sample buffer.

## Pre-embedding immunoperoxidase and immunogold electron microscopy (EM)

Following fixation in 0.05% glutaraldehyde and 4% PFA in 0.1 M phosphate buffer (PB), cell pellets were embedded in agar and cut onto 100 μm slices using a Vibratome. Slices were blocked with 20% normal goat serum and incubated with anti-Lamp1 (Invitrogen) or anti-LC3 (Cell Signalling) antibodies, followed by incubation with biotinylated anti-mouse and with goat anti-rabbit IgG coupled to 1.4 nm gold particles in 2% NGS/ 50 mM Tris-buffered saline. Slices were postfixed in 1% glutaraldehyde, and the gold particles were enlarged using the silver enhancement kit HQ-Silver from Nanoprobe. Slices were washed with ABC Elite Kit and peroxidase reaction was visualized by 3'3-diaminobenzidine. After washing, slices were postfixed with 0.5% OsO₄/ 1% uranyl acetate, dehydrated in a graded series of ethanol, treated with propylene oxide and embedded in Durcupan resin. Slices were cut into 60 nm ultrathin sections, counterstained with lead citrate and viewed in a Zeiss LEO 906 transmission electron microscope. Images were taken using the sharp-eye 2k CCD camera and processed with ImageSP.

## Flow cytometry analyses

Intracellular expression of HLA-DR in IFN-γ-stimulated HaCat cells, intracellular expression of pS6k and pAMPK in LCL cells and

intracellular expression of p62 in Ramos cells were determined as follows: $3 \times 10^5$ cells were first resuspended in PBS and stained for fixable viability dye (Invitrogen) for 20 min at 4 °C protected from light. Next, cells were fixed and permeabilized for 20 min using BD Cytofix/Cytoperm solution™ (BD), and then washed twice with 1× Perm/Wash Buffer™ (BD). Subsequently, cells were resuspended in 1× Perm/Wash Buffer and stained with conjugated anti-pS6K-Pacific Blue (Cell Signaling) for LCL cells, or with conjugated anti-HLA-DR (Invitrogen) for HaCat cells for 1 h at 4 °C or with conjugated anti-p62 (Abcam) for Ramos cells for 30 min at 4 °C. After washing two times, a secondary antibody anti-rabbit IgG-FITC (Invitrogen) was added to LCL cells and incubated at 4 °C for 25 min. Cells were then washed and acquired on a FACS Canto II (BD). Data analyses and calculation of the geometric mean fluorescence intensity (MFI) were performed using FlowJo™ 7.6.5 software (TreeStar Inc., Ashland, OR, USA). Gating strategy excluded doublets according to their FSC-H and FSC-A and dead cells, which were positive for fixable viability dye.

## ELISA

WT and LRBA-KO HaCat cells stably expressing GFP-LC3 or MP1-LC3 (target cells), and pre-treated with IFN-γ for 24 h to up-regulate MHC-II molecules were cultured with a MP1-specific CD4+ T cell clone (effector cells) from a HD (E:T = 10:5) for 20 h. Levels of TNF-α, IL-6, IFN-γ and MIP-1β were measured by ELISA in the collected supernatant as follows, 96-well flat-bottom plates (Corning Costar) were coated with 100 µl per well of 2 µg/ml of capture unlabeled mouse anti-human TNF-α (MABTECH), IL-6 (MABTECH,), IFN-γ (MAB-TECH), and MIP-1β (Thermo Fisher). Following overnight incubation at 4 °C, plates were incubated with blocking buffer (PBS with 0.05% Tween and 0.1% BSA) for 1 h at RT. Next, corresponding cytokine standard or serial dilutions of culture supernatant was added to each well and incubated for 2 h at RT. Biotin-conjugated detection antibody specific for each cytokine (MABTECH and Thermo Fisher) were added, and incubated for 1 h at RT. Streptavidin-HRP solution diluted 1:1000 in PBS was added and incubated for 1 h followed by incubation with TMB substrate (MABTECH) for 15 min protected from direct light. After stopping the reaction with 0.2 M H2SO4 the optical density was detected in an ELISA reader at 450 nm within 15 min. Cytokine levels of supernatants were determined according to the standard curve.

## In silico analyses

The LRBA protein interaction candidates were obtained from STRING and FUNCBase databases that provide known and predicted physical and functional protein-protein interactions. Only interactions with high confidence levels (>0.7) were selected from both databases for further validation. Network visualization was performed with the Cytoscape software provided for both databases.

## Modelling of the complex of LRBA and PIK3R4/PIK3C3

The structural modelling of the complex of LRBA and PIK3R4 was based on the finding that LRBA interacts via the WD40 domain with PIK3R4 (see results section). The 3D structure of the WD40 domain was modelled by homology modelling applying the structure of the homolog prokaryotic protein PkwA from *Thermomonospora curvata* (Brohawn et al, 2008). Sequence identity was 26% (39% shared similar residues). Since the first repeat sequence of the WD40-propeller of human LRBA is separated from the other WD40-repeats in sequence, this part of the domain was modelled based on a template of a single propeller and merged with the others, forming a full WD40 propeller domain. For the merging, a WD40 domain with an inserted propeller was used as a structural template (Wang et al, 2015)). For the structural modelling of the complex of PIK3R4 with PIK3C3 the homologous structures of the Phosphatidylinositol 3-kinase VPS34 in complex with serine/threonine-protein kinase VPS15 from Saccharomyces cerevisiae was applied. Human PIK3C and the template share a sequence identity of 36% (with 54% similar residues). Human PIK3R4 and VPS15 share a sequence identity of 33% (50% similar residues). The modelling was performed with the homology modelling software Prime (Schrödinger Suite 2018-1, LLC). For the model of the WD40 domain of PIK3R4 we applied the modelling server WDSPdb 2.0 that accurately identifies WD40 repeats and includes them for the correct assignment of the beta sheets in the structural model (Wang et al, 2015). The WD40 domain was then connected with the kinase domain of PIK3R4 and finally energy minimized using the OPLS3 force field provided by the Schrödinger software package. The modelling of the complex of PIK3R4/PIK3C3 and LRBA was performed by applying the docking web server HADDOCK (High Ambiguity Driven Biomolecular DOCKing, version 2.2, (van Zundert et al, 2016). As input constraints, the single exposed α-helix of the PIK3R4-WD40 domain (sequence motif: VGPSDD) and the smaller top surface of the WD40 domain of LRBA were defined as interacting regions. The top-ranked cluster with the highest negative Z-score value (-2.0) was selected for further analyses.

## Data and statistical analyses

Data analysis was performed without blinding and statistical significance was calculated with a non-parametric an unpaired Welch's *t* test, ratio paired Student *t* test, a one-way ANOVA with Tukey's multiple comparisons test or a two-way ANOVA with Bonferroni's multiple comparisons test using GraphPad Prism 6.0 software. A *P* value of <0.05 was considered statistically significant (*$P < 0.05$; **$P < 0.01$; ***$P < 0.001$; ****$P < 0.0001$).

# Data availability

This study includes no data deposited in external repositories.

The source data of this paper are collected in the following database record: biostudies:S-SCDT-10_1038-S44319-025-00504-7.

# Peer review information

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

## Acknowledgements

We thank Dr. Bernice Lo (National Institute of Allergy and Infectious Diseases, Bethesda, USA), Dr. Christian Behrends (University of Frankfurt, Germany) and Dr. Ian Gentle (University of Freiburg, Germany) for kindly providing us with plasmids. We thank Dr. Michael Moutschen (University of Liege, Belgium) and Prof. Manfred Kilimann (University of Gottingen, Germany) for kindly sharing with us the shLRBA HeLa cell line and the *Lrba*⁻/⁻ mice, respectively. In addition, we thank Jessica Rojas-Restrepo, Hanna Haberstroh, Türkan Güzel, Katja Malfertheiner, Vanessa Zeidler and Julika Neumann from our research group for their technical assistance during these past years, and to Dr. Virginia Andreani for her thorough review of this manuscript. We thank Dr. Marie Follo from the Light Core Facility Unit for her support and teaching of the microscopes. Schematic illustrations were created using Biorender licenses # QO24HLH0U1 and #KZ24HLGTRC. This study was funded by the Deutsche Forschungsgemeinschaft (DFG, German Research Foundation) under Project-ID 403222702 – SFB 1381, the Fritz Thyssen Foundation (grant number: 10.18.1.039MN), the EUCOR-Seed the Money grant (ACTIv), the Hans A. Krebs Medical Scientist Programme, Faculty of Medicine, University of Freiburg, and the Germany's Excellence Strategy (CIBSS-EXC-2189, Project ID 390939984) awarded to the Gámez-Díaz laboratory. The Grimbacher´s laboratory was supported by the Deutsche Forschungsgemeinschaft (DFG), grant numbers: GR1617/8-1, IMPATH-SFB (SFB1160/2), the Bundesministerium für Bildung und Forschung (BMBF) grant numbers: IFB/CCI: 01E01303 and E-med SysINFLAME: 012X1306F, 01GM1517C. The Kraft laboratory has received funding from Deutsche Forschungsgemeinschaft, DFG, Project-IDs 450216812, 409673687, SFB 1381 (Project ID 403222702), SFB 1177 (Project ID 259130777), under Germany's Excellence Strategy (CIBSS-EXC-2189- Project ID 390939984) and from the European Research Council (ERC) under the European Union's Horizon 2020 research and innovation programme under grant agreement No 769065. This work reflects only the authors' view and the European Union's Horizon 2020 research and innovation programme is not responsible for any use that may be made of the information it contains. Lighthouse Core Facility is funded in part by the Medical Faculty, University of Freiburg (Project Numbers 2021/B3-Fol) and the DFG (Project Number 450392965).

## Author contributions

**Elena Sindram**: Formal analysis; Validation; Investigation; Visualization; Methodology; Writing—original draft; Writing—review and editing. **Marie-Celine Deau**: Data curation; Formal analysis; Validation; Investigation; Visualization; Methodology; Writing—original draft; Writing—review and editing. **Laure-Anne Ligeon**: Conceptualization; Formal analysis; Investigation; Methodology; Writing—original draft. **Pablo Sanchez-Martin**: Formal analysis; Investigation; Visualization; Methodology. **Sigrun Nestel**: Investigation; Methodology. **Sophie Jung**: Formal analysis; Investigation; Methodology; Writing—original draft. **Stefanie Ruf**: Investigation; Methodology. **Pankaj Mishra**: Investigation; Methodology. **Michele Proietti**: Investigation; Methodology. **Stefan Günther**: Formal analysis; Investigation; Methodology. **Kathrin Thedieck**: Investigation; Methodology. **Eleni Roussa**: Conceptualization; Resources; Validation; Investigation; Methodology. **Angelika Rambold**: Resources; Formal analysis; Validation; Investigation; Methodology. **Christian Münz**: Conceptualization; Resources; Validation; Investigation. **Claudine Kraft**: Conceptualization; Resources; Data curation; Supervision; Funding acquisition; Validation. **Bodo Grimbacher**: Conceptualization; Resources; Software; Supervision; Funding acquisition; Writing—original draft; Project administration. **Laura Gámez-Díaz**: Conceptualization; Resources; Data curation; Software; Formal analysis; Supervision; Funding acquisition; Validation; Investigation; Visualization; Methodology; Writing—original draft; Project administration; Writing—review and editing.

Source data underlying figure panels in this paper may have individual authorship assigned. Where available, figure panel/source data authorship is listed in the following database record: biostudies:S-SCDT-10_1038-S44319-025-00504-7.

## Funding

## Disclosure and competing interests statement

The authors declare no competing interests.

# Expanded View Figures

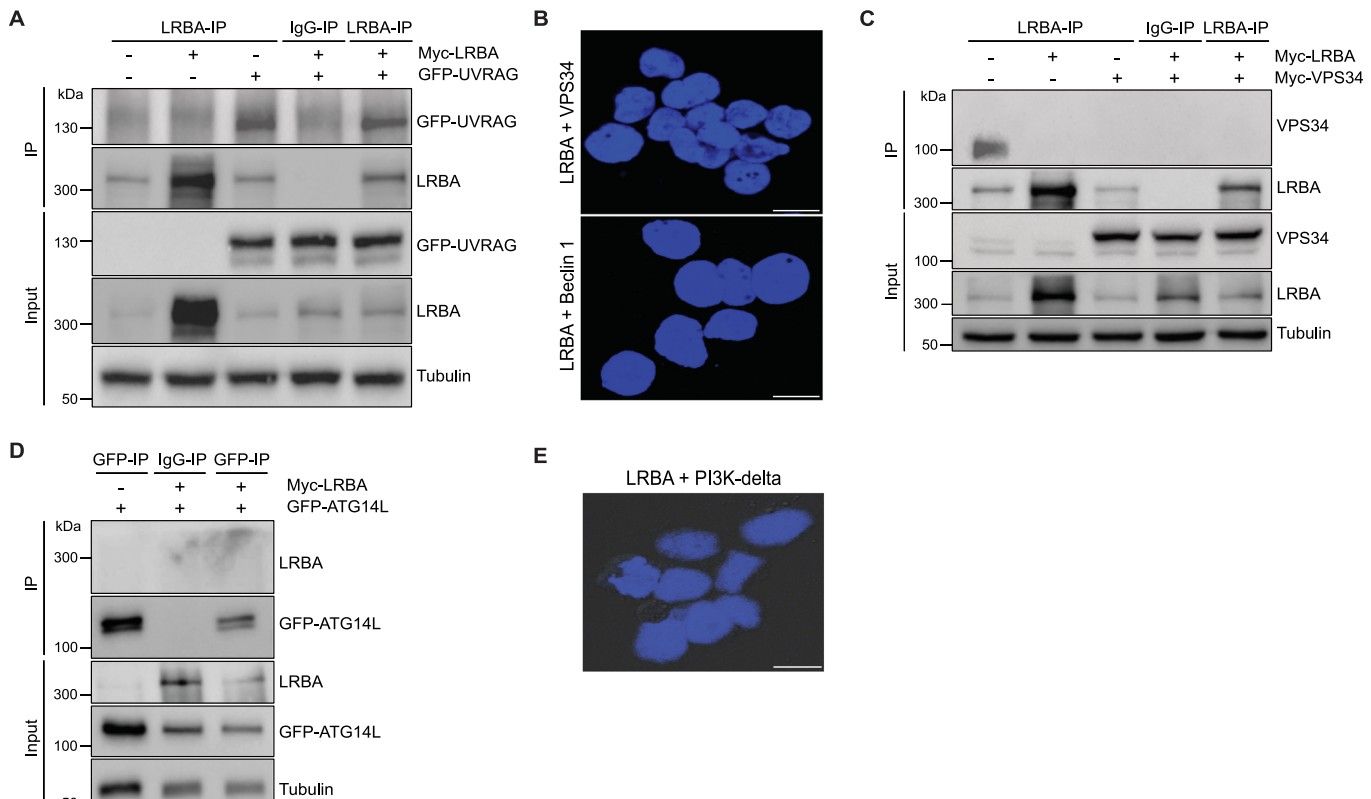

**Figure EV1.  LRBA interacts with UVRAG but not with other members of the PIKIII complex.**

(A) Co-IP analysis revealed an interaction between LRBA and UVRAG in HEK293T WT cells transfected with Myc-LRBA and GFP-UVRAG plasmids. Immunoprecipitation was performed with anti-LRBA and immunoblotted for GFP. (B) PLA of LRBA and VPS34 (top) or Beclin-1 (bottom) in LCL cells from a healthy donor (HD) under resting conditions, revealing a lack of proximity. Nuclear DAPI staining is shown in blue. Scale bar=20 μm. (C) HEK293T WT cells were transfected with Myc-LRBA plasmid and/or Myc-VPS34/His-PIK3R4 plasmid (PIK3R4 blots are shown in Fig. 1E). Immunoprecipitation was performed with anti-LRBA and immunoblotted for VPS34. VPS34 and PIK3R4 were detected in the same experiment, therefore images of LRBA input and IP are similar as in Fig. 1E. (D) HEK293T WT cells were transfected with Myc-LRBA plasmid and GFP-ATG14L plasmid. Immunoprecipitation was performed with anti-LRBA and immunoblotted for GFP. (E) PLA showing the absence of proximity of LRBA with PI3Kdelta in LCL cells from a HD under resting conditions. Nuclear DAPI staining is shown in blue. Scale bar = 20 μm.

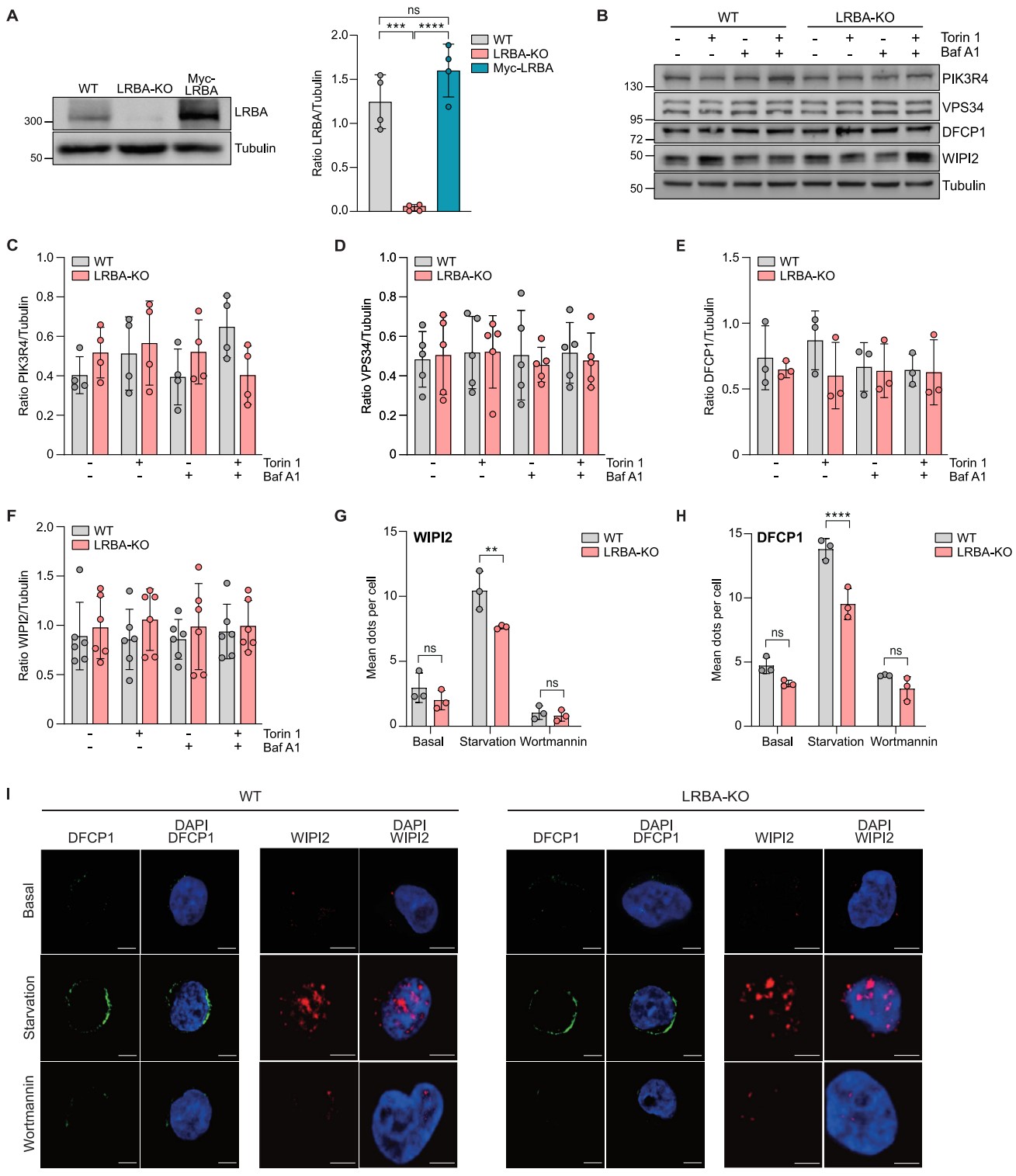

◀ **Figure EV2. Diminished DFCP1/WIPI2 punctae under starvation.**

(A) Representative immunoblot (left) and densitometry analyses (right) of LRBA expression in relation to Tubulin in WT (grey), LRBA-KO (red) and Myc-LRBA reconstituted (teal) HEK293T cells. Each dot represent one blot while bars represent the mean ± SD of $n = 4$ independent biological replicates. (B–F) Loss of LRBA does not affect the protein expression of PIK3R4, VPS34, DFCP-1 and WIPI-2. Representative immunoblot (B) and densitometry analysis of the protein levels of (C) PIK3R4, (D) VPS34, (E) DFCP1 and (F) WIPI2 in relation to Tubulin in the presence and absence of Torin 1, with or without Bafilomycin A1, for WT and LRBA-KO HEK293T cells. Each dot represents the densitometry analysis of one blot while bars represent the mean ± SD from $n = 3$–6 independent biological replicates. (G–I) (G) Quantification of WIPI2 and (H) DFCP1 expression in WT (grey) and LRBA-KO (red) HEK293T cells cultured for 1 h with EBSS or 100 nM Wortmannin. Each dot represents the mean of one experiment while bars represent the mean ± SD of $n = 3$ independent biological replicates. Total cells analyzed (WIPI2/DFCP1) for WT EBSS: 14/60 cells and KO = 28/42 cells. For WT=Wortmannin 29/57 cells and KO = 26/83 cells. (I) Representative confocal microscopy images of DFCP1 (green) and WIPI2 (red) signal upon staining with anti-DFCP1 and anti-WIPI2 antibodies. Scale bar=5 μm. Statistical analyses of (A) was performed using a one-way ANOVA with Tukey's multiple comparisons test and for (C–H) a two-way ANOVA with Bonferroni's multiple comparisons test, **$P < 0.01$ ((G): $P = 0.0052$), ***$P < 0.001$ ((A): $P = 0.0006$), ****$P < 0.0001$.

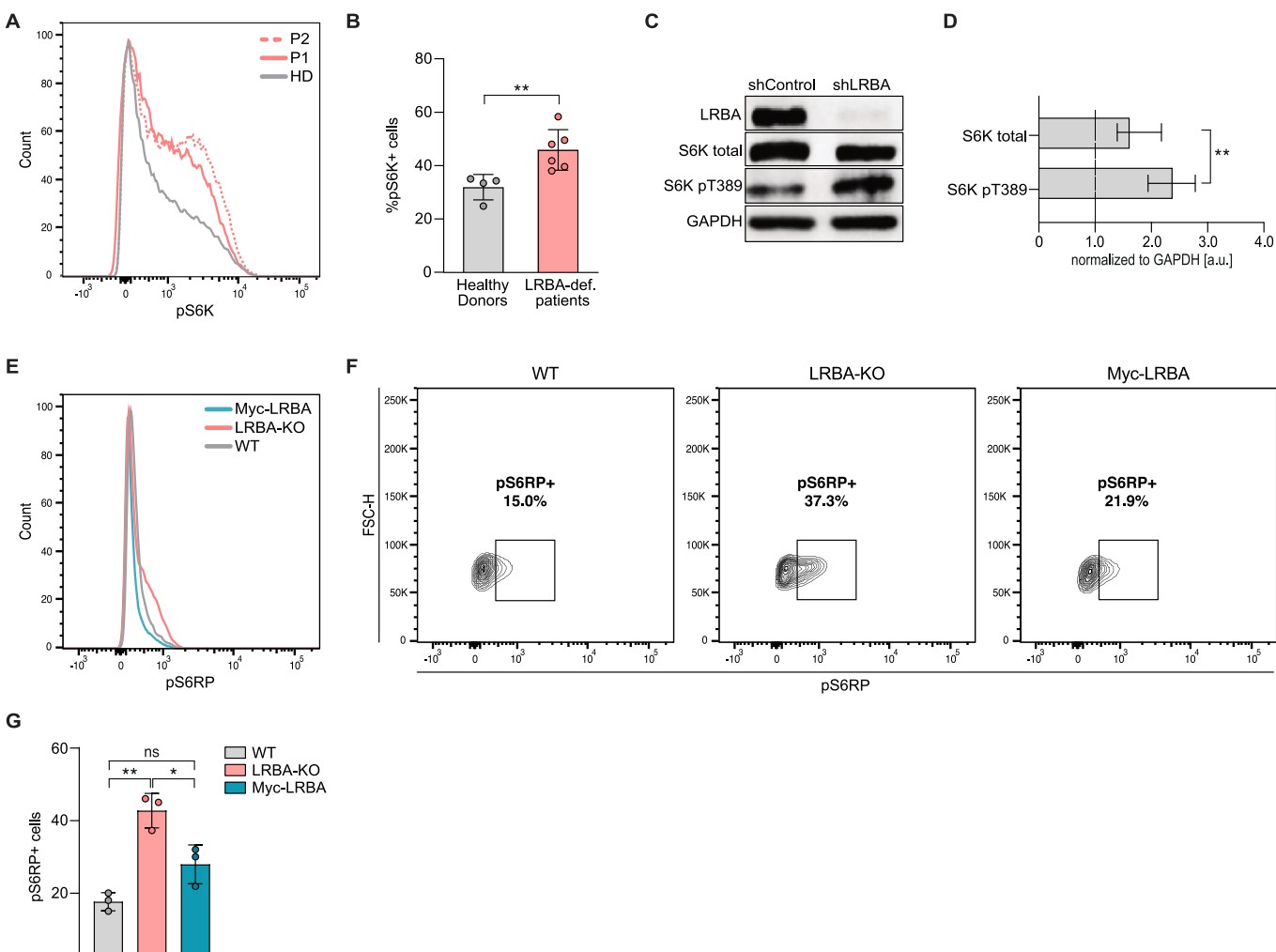

**Figure EV3. Enhanced mTOR signaling in LRBA-deficient cells.**

(A) Representative histogram of pS6K-MFI detected by flow cytometry in LCL from HD (grey) and LRBA-deficient patients (red) at basal levels. (B) Bar graphs representing the percentage of cells positive for pS6K in HD (grey) and two LRBA-deficient patients (red). Each dot represents the mean of $n = 2$ technical replicates while bars present the mean ± SD of $n = 4$ independent biological replicates. (C) Representative immunoblot analyses of LRBA, S6K total, S6K pT389 and GAPDH performed in shControl and shLRBA-HeLa cells at basal conditions. (D) Bar graphs represent densitometry analyses of S6K total and S6K pT389 expression from HeLa cells. Quantifications of protein expression were previously normalized to GAPDH and to shControl (dotted line). The bars represent the means ± SD from $n = 4$ independent biological replicates. (E–G) pS6RP levels were rescued after LRBA reconstitution. (E) Representative histogram of pS6RP-MFI (F) representative dot plot of p6RP+ cells, and (G) percentage of cells positive for pS6RP that was detected by flow cytometry in WT (grey), LRBA-KO (red) and Myc-LRBA (teal) at basal levels. Each dot represents the mean of $n = 2$ technical replicates while bars present the mean ± SD of $n = 3$ independent biological replicates. Statistical analyses of (B, D) was performed using a unpaired Welch's $t$ test and for (G) a one-way ANOVA with Tukey's multiple comparisons test, *$P < 0.05$ ((G): $P = 0.0144$), **$P < 0.01$ ((B): $P = 0.0073$), ((G): $P = 0.001$).

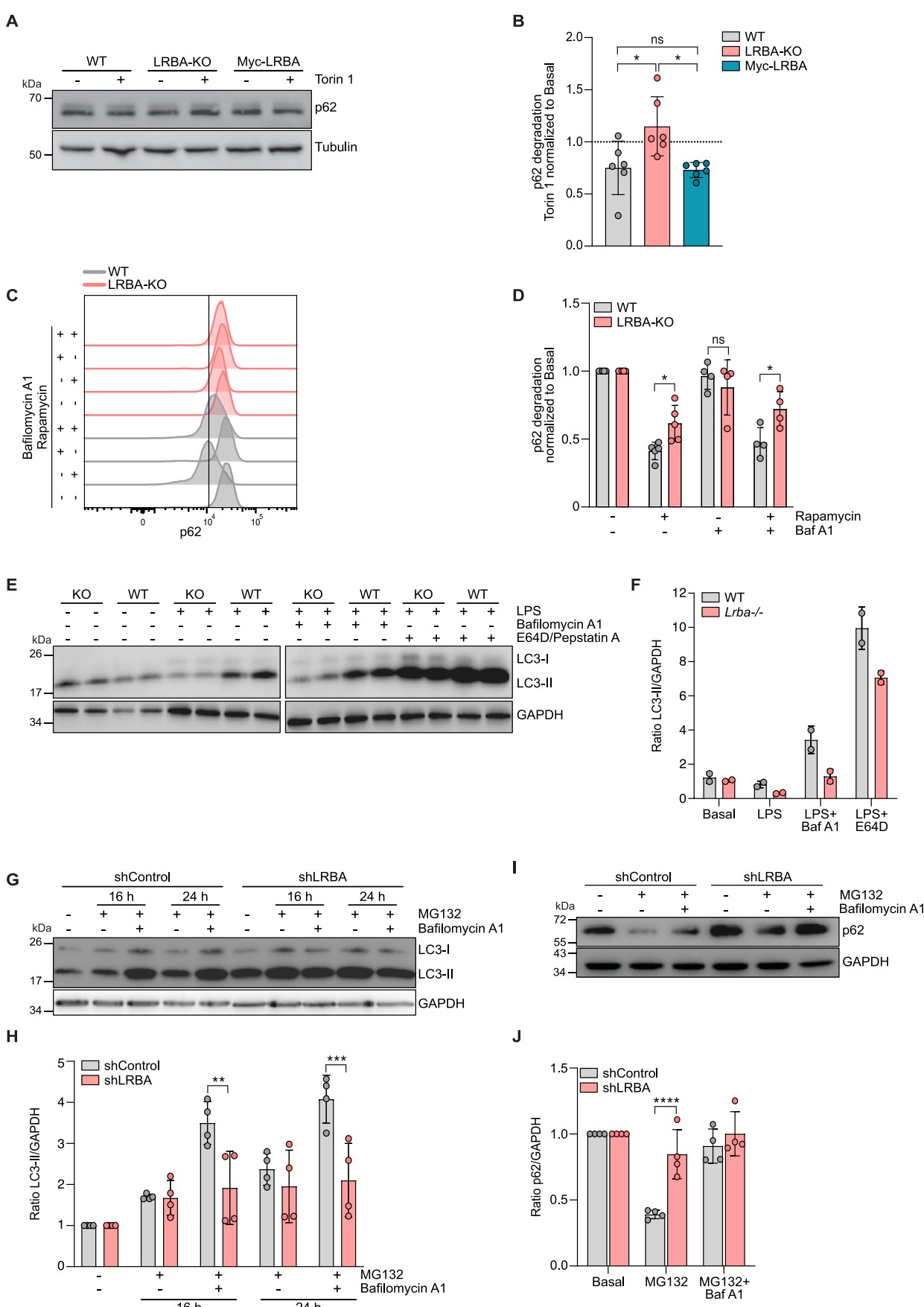

◀ **Figure EV4. Abnormal autophagy flux in shLRBA HeLa cells and B cells from *Lrba−/−* mice.**

(A) Representative immunoblot analyses of p62 degradation in WT, LRBA-KO and Myc-LRBA HEK293T cells at basal conditions or after Torin 1 stimulation. (B) Densitometry analysis of (A) of WT (grey), LRBA-KO (red) and Myc-LRBA (teal) HEK293T cells. Each dot represents the densitometry analysis of one blot while bars represent the mean ± SD from $n = 6$ independent biological replicates. (C, D) Representative histograms of p62 degradation in WT (grey) and LRBA-KO (red) Ramos B cells in the presence and absence of Rapamycin, with or without Bafilomycin A1 and (D) fold change of p62 degradation normalized to basal levels. Each dot represents the fold change of one technical replicate while bars represent the mean ± SD from $n = 4–5$ independent biological replicates. (E, F) Reduced autophagy flux in splenic naive B cells from *Lrba−/−* mice. (E) Representative immunoblot analyses of LC3-I and LC3-II processing and Tubulin in isolated splenocytes from WT and Lrba$^{-/-}$ mice at day 0 or upon stimulation for 3 days with 20 μg/ml LPS alone or in the presence of 100 nM Bafilomycin A1 or protease inhibitors E64D and Pepstatin. (F) Densitometry analyses of LC3-II expression relative to Tubulin of splenic murine naive B cells from WT (grey) and *Lrba−/−* (red) mice. Each dot represents the densitometry analysis of one blot while bars represent the mean ± SD from $n = 2$ independent biological replicates. (G) Representative immunoblot analyses of the processing of endogenous unconjugated LC3-I to lipid-conjugated LC3-II in shControl and shLRBA HeLa cells at resting conditions or after 16 h or 24 h of MG132 alone or in the presence of 100 nM Bafilomycin A1. (H) Densitometry analyses of LC3-II expression relative to GAPDH of shControl (grey) and shLRBA (red) HeLa cells. Each dot represents the densitometry analysis of one blot while bars represent the mean ± SD from $n = 4$ independent biological replicates. (I) Representative immunoblot analyses of p62 and GAPDH in shControl and shLRBA HeLa cells at resting conditions or after 16 h treatment with MG132 alone or along with 100 nM Bafilomycin A1. (J) Densitometry analyses of p62 relative to GAPDH and normalized to basal conditions of shControl (grey) and shLRBA (red) HeLa cells. Each dot represents the densitometry analysis of one blot while bars represent the mean ± SD from $n = 4$ independent biological replicates. Statistical analysis for (B) was performed using a one-way ANOVA with Tukey's multiple comparisons test and for (D, H, I) a two-way ANOVA with Bonferroni's multiple comparisons test, *$P < 0.05$ (B: $P = 0.02$ WT vs KO and $P = 0.0146$ KO vs Myc-LRBA), (D: $P = 0.0237$ Rapamycin and $P = 0.0132$ Rapamycin+Bafilomycin A1), **$P < 0.01$ ((H): $P = 0.0027$), ***$P < 0.001$ ((H): $P = 0.0002$), ****$P < 0.0001$.

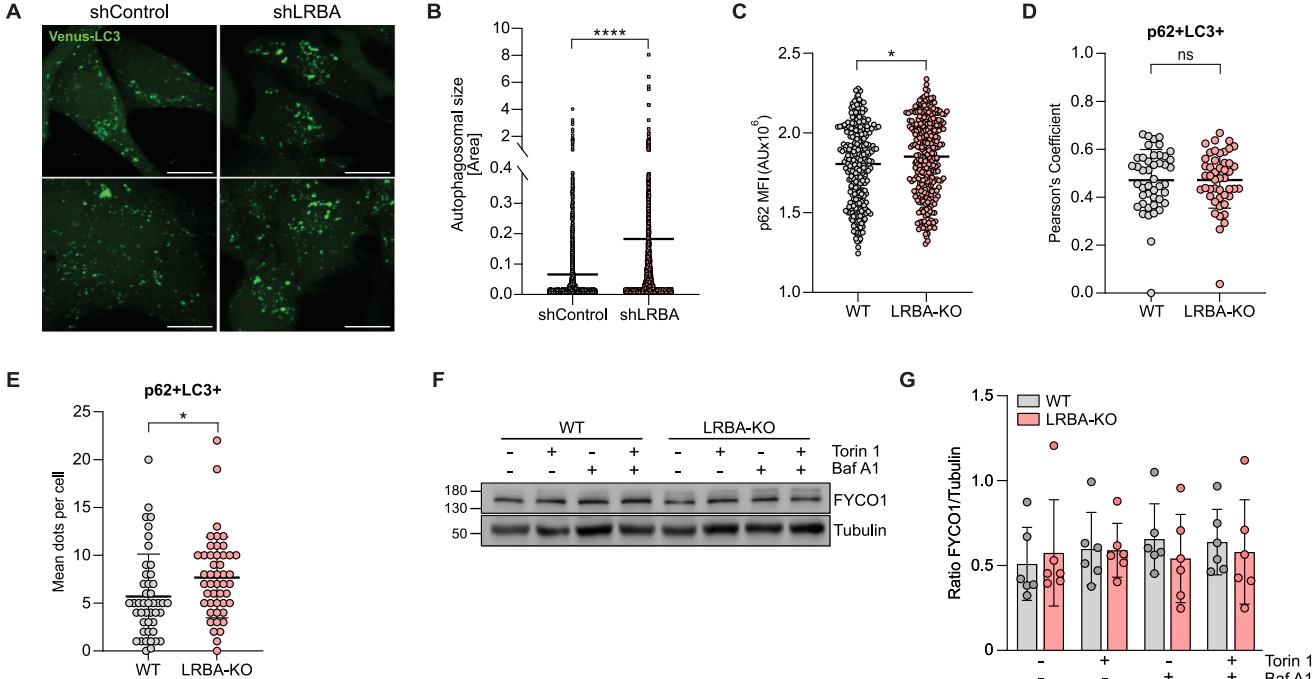

**Figure EV5. Loss of LRBA leads to enlarged autophagosomes.**

(A, B) shControl and shLRBA HeLa cells were transfected with GFP-LC3 plasmid and incubated for 2 h with 100 nM of Bafilomycin A1 for microscopy evaluation. (A) Representative confocal microscopy images of LC3-transfected shControl and shLRBA HeLa cells (green). Scale bar = 10 μm. (B) Size of GFP-LC3 punctae in shControl (grey) and shLRBA (red) HeLa cells was determined from binary images using the analyses particle module of FIJI, with a particle size from 0 to 20 μm². Each dot represents an autophagosome from n = 3 fields across n = 3 independent biological replicates. Total autophagosomes for shControl = 961 and shLRBA= 1249. (C–E) Scatter plots showing the (C) p62 MFI in p62+LC3+ dots (D) co-localization of LC3 and p62 and (E) number of LC3+p62+ dots in WT (grey) and LRBA-KO (red) HaCat cells. Each dot for (C) represents one autophagosome from n = 3 independent biological replicates with a total of n = 262 (WT) and n = 225 (LRBA-KO) analyzed and for (D, E) one cell from n = 3 independent biological replicates with a total of at least 43 cells analyzed per condition, mean ± SD shown by the black line and error bars. (F) Representative immunoblot analyses and (G) densitometry analysis of FYCO1 and Tubulin expression upon basal conditions and stimulation with Torin 1 in presence or absence of Bafilomycin A1. Each dot represents the densitometry analysis of one blot while bars represent the mean ± SD from n = 6 independent biological replicates. Statistical analysis for (B–E) was performed using an unpaired Welch's t test and for (G) two-way ANOVA with Bonferroni's multiple comparisons test, *P < 0.05 ((C): P = 0.0258), ((E): P = 0.0304), ****P < 0.0001.

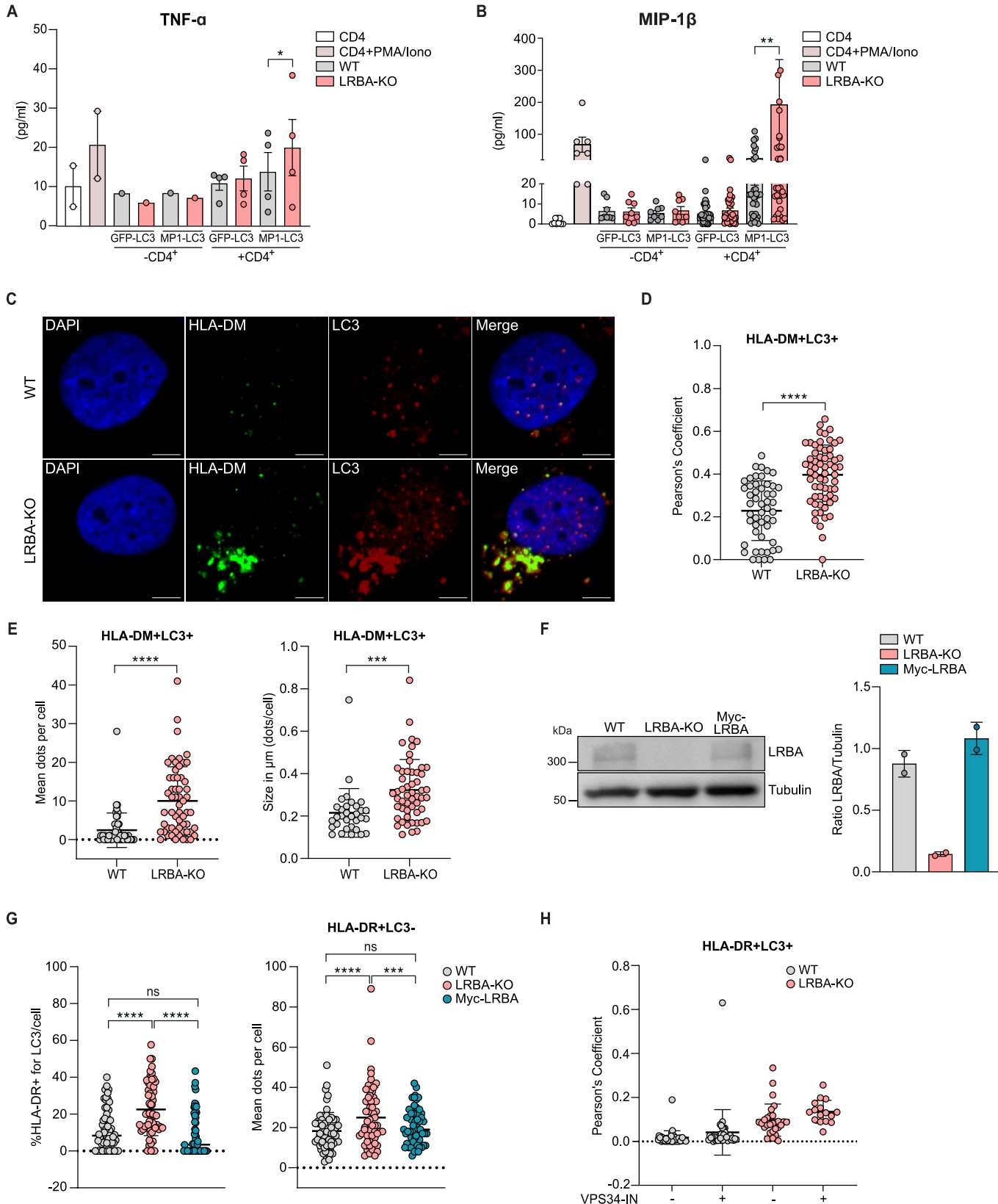

**Figure EV6. Loss of LRBA leads to increased TNF-α and MP-1β release *via* autophagy.**

(A, B) WT and LRBA-KO HaCat cells stably expressing GFP-LC3 or MP1-LC3 (target cells), and pre-treated with IFN-γ for 24 h to up-regulate MHC-II were cultured with a MP1-specific CD4⁺ T cell clone (effector cells) from a HD for 20 h. Following incubation, TNF-α and MIP-1β were measured in the culture supernatants by ELISA. Each dot represents one technical replicate while bars represent the mean ± SD of (A) TNF-α ($n = 4$ independent biological replicates for HaCat GFP/MP-1-LC3 cells in presence of CD4+ T cells, whereas $n = 1$ for HaCat GFP/MP-1-LC3 without CD4+ T cells, and $n = 2$ for CD4+ cells alone) and (B) MIP-1β ($n = 4$ independent biological replicates) secretion in WT (grey) and LRBA-KO (red) HaCat cells (C) Representative confocal microscopy images of WT and LRBA-KO HaCat cells stimulated overnight with IFN-γ and treated with 20 μM chloroquine for 6 h. Fixed cells were stained for HLA-DM (green), LC3 (red) and DAPI (blue) for the nuclei. Scale bar = 5 μm. (D, E) Scatter plots showing the (D) co-localization of LC3 and HLA-DM and (E) the number of dots per cell (left) and size (right) of vesicles positive for HLA-DM and LC3 in WT (grey) and LRBA-KO (red) HaCat cells. Each dot represents one cell from $n = 3$ independent biological replicates with a total of at least 35 cells analyzed per condition, mean ± SD shown by the black line and error bars. (F) Representative immunoblot and densitometry analyses of LRBA expression in WT (grey), LRBA-KO (red) and Myc-LRBA (teal) reconstituted HaCat cells. Each dot represents the densitometry analysis of one blot while bars represent the mean ± SD from $n = 2$ independent biological replicates (G) Scatter plots showing co-localization of HLA-DR and LC3 (left) and number of HLA-DR+LC3- dots (right) in WT (grey), LRBA-KO (red) and Myc-LRBA (teal) reconstituted HaCat cells. Each dot represents one cell from $n = 3$ independent biological replicates with a total of at least 30 cells analyzed per condition, mean ± SD shown by the black line and error bars. (H) Scatter plots showing the co-localization of HLA-DR and LC3 in unstimulated WT (grey) and KO (red) HaCat cells with or without VPS34 inhibitor (0.1 μM) treatment. Each dot represents one cell from $n = 2$ biological replicates with a total of at least 30 cells analyzed per condition, mean ± SD shown by the black line and error bars. Statistical analyses of (A, B) was performed using a ratio paired Student *t* test, for (D, E) a unpaired Welch's *t* test, for (G) a one-way ANOVA with Tukey's multiple comparisons test and for (H) and a two-way ANOVA with Bonferroni's multiple comparisons, *$P < 0.05$ ((A): $P = 0.0410$), **$P < 0.01$ ((B): $P = 0.0014$), ***$P < 0.0001$ ((E): $P = 0.0003$), ((G): $P = 0.0004$), ****$P < 0.0001$.

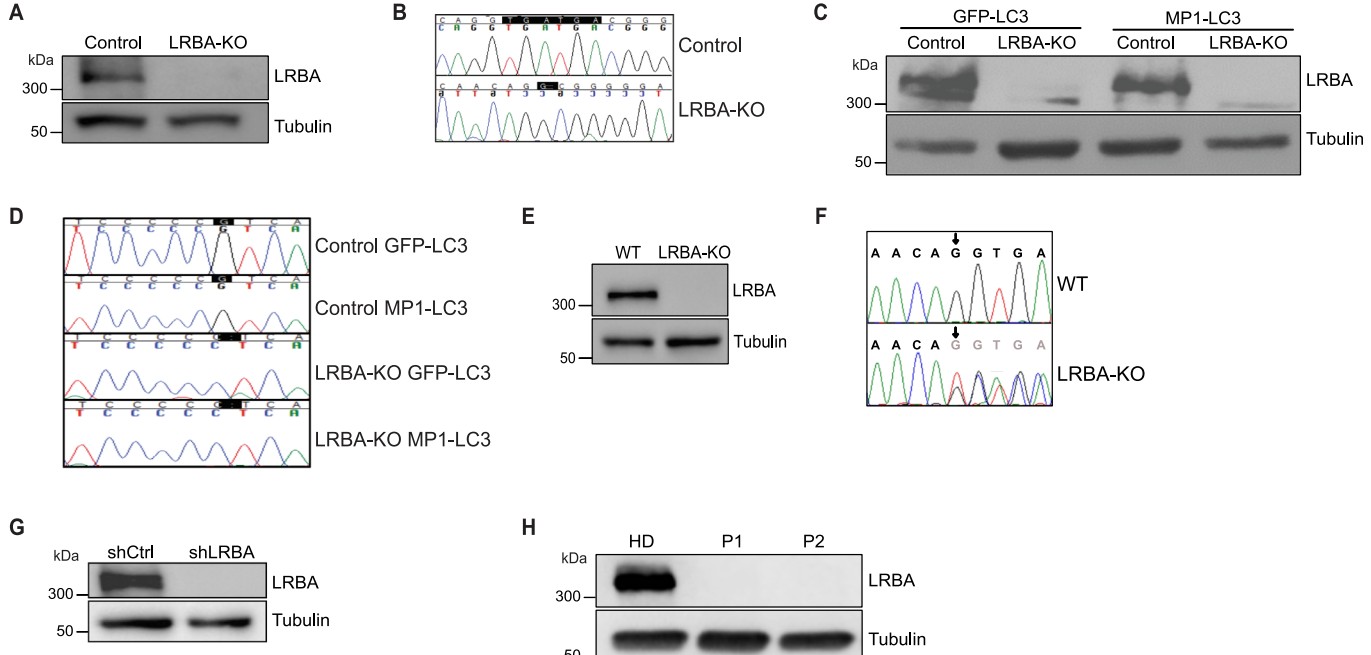

**Figure EV7. Generation of LRBA-deficient cell lines.**

(A, C, E) Immunoblot analyses of LRBA protein expression in WT and LRBA-KO (A) HEK293T, (C) HaCaT and (E) Ramos cells. (B, D, F) Sequencing analyses of *LRBA* exon 2 cells showing successful depletion of LRBA using the CRISPR-Cas9 system in (B) HEK293T, (D) HaCaT and (F) Ramos cells. (G) Immunoblot analyses of LRBA protein expression in HeLa cells shcontrol and shLRBA HeLa cells. (H) Immunoblot analyses of LRBA protein expression in LCL cells from a HD and two LRBA-deficient patients.

