## [Peer Review File · EMBO Reports]

LRBA deficiency impairs autophagy, enhancing antigen presentation and contributing to T-cell dysregulation

Elena Sindram, Marie-Celine Deau, Laure-Anne Ligeon, Pablo Sanchez-Martin, Sigrun Nestel, Sophie Jung, Stefanie Ruf, Pankaj Mishra, Michele Proietti, Stefan Günther, Kathrin Thedieck, Eleni Roussa, Angelika Rambold, Christian Münz, Claudine Kraft, Bodo Grimbacher, and Laura Gámez-Díaz

Corresponding author(s): Laura Gámez-Díaz (laura.gamez@uniklinik-freiburg.de) , Bodo Grimbacher (bodo.grimbacher@uniklinik-freiburg.de)

Review Timeline:

Submission Date:	16th Dec 24
Editorial Decision:	13th Mar 25
Revision Received:	15th Apr 25
Editorial Decision:	2nd May 25
Revision Received:	9th May 25
Accepted:	26th May 25

Transaction Report: Please note that the manuscript was transferred from another journal where it was originally reviewed. Since the original reviews are not subject to EMBO's transparent review process policy, they cannot be published.

Dear Laura,

Thank you for the submission of your research manuscript to our journal. I sincerely apologize for the unusual delay in handling your manuscript.

Let me shortly summarize the history of your manuscript. Your manuscript has undergone peer review and revision in response to the referee comments at another journal and we have taken the referee reports you had obtained on the revised version into account in our editorial evaluation at EMBO Reports. Former referee 1 and 3 had considered your response to the referee concerns adequate, but referee 2 had raised remaining concerns regarding the contribution of aberrant autophagy regulation in LRBA deficiency for the hyperinflammatory disease phenotype. The referee noted that the observation of aberrant autophagy regulation was overall convincing after the revision but how important it is for the autoinflammatory phenotype and its relative contribution in comparison to other mechanisms, such as defective CTLA-4 homeostasis, would need further clarification. While this concern is without doubt relevant, we overall felt that - for publication at EMBO Reports - it can be addressed by toning down conclusions and by discussing this limitation in the manuscript.

You informed me that you had meanwhile further revised your manuscript and performed additional experiments before submitting it to our journal, such as the analysis of endogenous expression levels of FYCO1, DFCP1, WIPI2, PIK3R4 and VPS34 proteins or the endogenous co-IP of LRBA and PIK3R4 in WT and LRBA-KO HEK293T cell lines und basal conditions. You had further added data on HLA-DR localization and expression, shown in Figure 6D-I.

I have analysed the new data myself and in addition asked former referee 3 to check the newly added data. The referee has meanwhile replied and considered the newly added data convincing. That said, the referee also commented on a few points that should be addressed in the discussion.

1) The referee commented on the data shown in Figure 6: "The study posits that impaired autophagosome-lysosome fusion in LRBA-KO cells leads to increased fusion with MIIC vesicles, enhancing antigen presentation (Fig. 6). This is puzzling, as defective autophagy typically reduces antigen delivery to MHC-II compartments. Can the authors provide direct evidence (e.g., co-localization of autophagosomal markers like LC3 with MIIC markers beyond HLA-DR, or live-cell imaging of vesicle dynamics) to demonstrate this rerouting? Additionally, how do enlarged autophagosomes (Fig. 4) contribute to this process-do they contain more antigenic cargo, or is their fusion efficiency with MIICs altered?"

And along the same lines: "The interaction between LRBA and FYCO1 is well-documented (Fig. 5), and reduced autophagosome velocity in shLRBA cells is compelling (Fig. 5D-F). However, the manuscript does not connect FYCO1 dysfunction to enhanced antigen presentation. Does FYCO1 knockdown phenocopy LRBA deficiency in MHC-II presentation assays? This could clarify whether impaired autophagosome transport directly contributes to MIIC accumulation."

These two concerns should either be addressed with additional stainings for MIIC markers (Fig 6), or at least be discussed in the manuscript.

2) The referee further commented on the statistical analysis, and this concern should be addressed by being specific in the figure legends about sample size and statistical test used. If the data rely on 2 replicates, please do not perform statistical analysis. In all cases, the individual datapoints have to be shown in addition to the mean or average, if applicable: "Some experiments (e.g., Fig. 2G, n=2) have low sample sizes, and it's unclear whether "n" represents biological or technical replicates. The authors should specify this in the figure legends and, where possible, increase replicates to enhance statistical power. Additionally, the use of multiple statistical tests (Mann-Whitney, Welch's t-test, Two-Way ANOVA) without correction for multiple comparisons could inflate type I errors-consider applying a Bonferroni or FDR adjustment."

3) Please also discuss a third concern brought forward, i.e., that the data rely to quite some extent on non-immune cell lines, such as HEK293T and HaCat cells, with only limited evidence provided for autophagy flux and antigen presentation in immune cells, such as LRBA KO B-cells (Figure 4G, S4G-H).

Given the positive evaluation by former referee 3, we will proceed with publication of your article, pending that the concerns listed above are addressed, at least in textual form.

I list the general formatting guidelines below, but already list here some specific points that will require your attention. Once the revised manuscript has been submitted, it will undergo an editorial quality check again. All minor issues that will come up at this point, can then be addressed in a second step.

Specific points:

1) Please supply the movie legends as README.txt file and zip it with the movie. The zip file is uploaded.

- 2) Please provide the exact p-values in the figure legends (unless $p < 0.0001$) and please do not perform statistical analysis based on 2 repeats.
- 3) Regarding the Author Contributions, we now use CRediT to specify the contributions of each author in the journal submission system. Therefore, please remove the Author Contributions from the manuscript file and make sure that the author contributions in our online manuscript tracking system are correct and up-to-date. The information you specified in the system will be automatically retrieved and typeset into the article. You can enter additional information in the free text box provided, if you wish.
- 4) Please update the 'Conflict of interest' paragraph to our new 'Disclosure and competing interests statement'. For more information see <https://www.embopress.org/page/journal/14693178/authorguide#conflictsofinterest>
- 5) The Data Availability statement should only refer to data deposited in public databases. Please remove the current statement and replace it with: "This study includes no data deposited in external repositories."
- 6) References: et al should be used if there are more than 10 authors, i.e., the first 10 authors are listed followed by et al.
- 7) The Expanded View Figure Legends come directly after the main Figure Legends.
- 8) The nomenclature for Expanded View content is: Movie EV#, Figure EV#.
- 9) Materials and Methods are called Methods and we need a Reagent and Tools Table (see point 12 below).
- 10) Please provide the Author Checklist (see below).

GENERAL GUIDELINES:

- 1) a .docx formatted version of the manuscript text (including legends for main figures, EV figures and tables). Please make sure that the changes are highlighted to be clearly visible.
- 2) individual production quality figure files as .eps, .tif, .jpg (one file per figure). Please download our Figure Preparation Guidelines (figure preparation pdf) from our Author Guidelines pages <https://www.embopress.org/page/journal/14693178/authorguide> for more info on how to prepare your figures.
- 3) a .docx formatted letter INCLUDING the reviewers' reports and your detailed point-by-point responses to their comments. As part of the EMBO Press transparent editorial process, the point-by-point response is part of the Review Process File (RPF), which will be published alongside your paper.
- 4) a complete author checklist, which you can download from our author guidelines (<<https://www.embopress.org/page/journal/14693178/authorguide>>). Please insert information in the checklist that is also reflected in the manuscript. The completed author checklist will also be part of the RPF.
- 5) Please note that all corresponding authors are required to supply an ORCID ID for their name upon submission of a revised manuscript (<<https://orcid.org/>>). Please find instructions on how to link your ORCID ID to your account in our manuscript tracking system in our Author guidelines (<<https://www.embopress.org/page/journal/14693178/authorguide#authorshipguidelines>>)
- 6) We replaced Supplementary Information with Expanded View (EV) Figures and Tables that are collapsible/expandable online. A maximum of 5 EV Figures can be typeset. EV Figures should be cited as 'Figure EV1, Figure EV2' etc... in the text and their respective legends should be included in the main text after the legends of regular figures.
 - For the figures that you do NOT wish to display as Expanded View figures, they should be bundled together with their legends in a single PDF file called *Appendix*, which should start with a short Table of Content. Appendix figures should be referred to in the main text as: "Appendix Figure S1, Appendix Figure S2" etc. See detailed instructions regarding expanded view here: <<https://www.embopress.org/page/journal/14693178/authorguide#expandedview>>
 - Additional Tables/Datasets should be labeled and referred to as Table EV1, Dataset EV1, etc. Legends have to be provided in a separate tab in case of .xls files. Alternatively, the legend can be supplied as a separate text file (README) and zipped together with the Table/Dataset file.

7) Please include a dedicated "Data Availability" section at the end of the Methods (suggested wording: "The [structural coordinates | microarray | mass spectrometry] data from this publication have been deposited to the [name of the database] database [URL] and assigned the identifier [accession | permalink | hashtag]."). Should this not apply, this should still be stated as "This study includes no data deposited in external repositories."

Additional information on source data and instruction on how to label the files are available
<<https://www.embopress.org/page/journal/14693178/authorguide#sourcedata>>.

10) Figure legends and data quantification:
The following points must be specified in each figure legend:

- the name of the statistical test used to generate error bars and P values,
 - the number (n) of independent experiments (please specify technical or biological replicates) underlying each data point,
 - the nature of the bars and error bars (s.d., s.e.m.)
- If the data are obtained from n {less than or equal to} 5, show the individual data points in addition to the SD or SEM.
 - If the data are obtained from n {less than or equal to} 2, use scatter blots showing the individual data points.

See also the guidelines for figure legend preparation:
<https://www.embopress.org/page/journal/14693178/authorguide#figureformat>

11) Our journal encourages inclusion of *data citations in the reference list* to directly cite datasets that were re-used and obtained from public databases. Data citations in the article text are distinct from normal bibliographical citations and should directly link to the database records from which the data can be accessed. In the main text, data citations are formatted as follows: "Data ref: Smith et al, 2001" or "Data ref: NCBI Sequence Read Archive PRJNA342805, 2017". In the Reference list, data citations must be labeled with "[DATASET]". A data reference must provide the database name, accession number/identifiers and a resolvable link to the landing page from which the data can be accessed at the end of the reference. Further instructions are available at <<https://www.embopress.org/page/journal/14693178/authorguide#referencesformat>>.

12) All Materials and Methods need to be described in the main text using our 'Structured Methods' format. According to this format, the Methods section includes a Reagents and Tools Table (listing key reagents, experimental models, software and relevant equipment and including their sources and relevant identifiers) followed by a Methods and Protocols section describing the methods, ideally using a step-by-step protocol format. The aim is to facilitate adoption of the methodologies across labs. Please download and fill our Reagents and Tools Table template (.docx), which you can find in our author guidelines: <https://www.embopress.org/page/journal/14693178/authorguide#structuredmethods>.
When submitting your revised manuscript, please do not include the Reagents and Tools Table in the Methods section of the manuscript but upload it as a separate file choosing the file type "Reagent Table".
An example of a Method paper with Structured Methods can be found here:
<https://www.embopress.org/doi/10.15252/msb.20178071>.

13) As part of the EMBO publication's Transparent Editorial Process, EMBO Reports publishes online a Review Process File to accompany accepted manuscripts. This File will be published in conjunction with your paper and will include the referee reports, your point-by-point response and all pertinent correspondence relating to the manuscript.

Kind regards,

Martina

Point-by-point replies to Editor and Reviewer's comments:

Please find below the Reviewers' comments in black and the authors' responses in blue.

Editor:

These two concerns should either be addressed with additional staining for MIIC markers (Fig 6), or at least be discussed in the manuscript.

Reviewer:

1. The study posits that impaired autophagosome-lysosome fusion in LRBA-KO cells leads to increased fusion with MIIC vesicles, enhancing antigen presentation (Fig. 6). This is puzzling, as defective autophagy typically reduces antigen delivery to MHC-II compartments. Can the authors provide direct evidence (e.g., co-localization of autophagosomal markers like LC3 with MIIC markers beyond HLA-DR, or live-cell imaging of vesicle dynamics) to demonstrate this rerouting? Additionally, how do enlarged autophagosomes contribute to this process (increased fusion with MIIC vesicles, enhancing antigen presentation) -do they contain more antigenic cargo, or is their fusion efficiency with MIICs altered?"

We thank the reviewer for this valuable comment. Following the suggestion, we investigated the co-localization of the autophagosomal marker LC3 with an additional MIIC marker, HLA-DM, in MP1-LC3 overexpressed wild type (WT) and LRBA-KO HaCat cells upon stimulation with IFN- γ and chloroquine. Our confocal microscopy analysis revealed **increased co-localization of LC3 with HLA-DM, as well as enlarged LC3+HLA-DM+ vesicles in LRBA-KO cells**. These findings corroborate our previous results showing an increase in the number and size of LC3+HLA-DR+ vesicles, further supporting the **augmented number of MIIC vesicles in LRBA-deficient cells**. This data suggests that the impaired autophagosome-lysosome fusion in LRBA-KO cells leads to the rerouting of autophagosomes towards MIIC vesicles. This new data set is now included in the revised version of our manuscript as Expanded View Figure 6, C-E, and described in the results section lines: 296 – 297.

Using the same study model, we observed similar co-localization of LC3 with p62 in both WT and LRBA-KO HaCat cells upon autophagy induction. **However, LRBA-KO cells exhibited a higher number of LC3+p62+ vesicles, which were also significantly larger, compared to WT cells**. This is consistent with the accumulation of enlarged autophagosomes previously observed in LRBA-KO HEK293T cells and shLRBA HeLa cells. **Notably, the**

enlarged autophagosomes were associated with increased p62 cargo content. These observations are now included in the Results section (lanes: 232-238) and in Figure 4, G –I, and Expanded View Figure 5, C-E.

Taken together, our data suggest that impaired fusion of autophagosomes with lysosomes in LRBA-KO cells leads to an accumulation of autophagosomes (LC3+) that are rerouted to MIIC vesicles. Furthermore, we speculate that the larger autophagosomes, observed more abundant in LRBA-KO cells, with their increased cargo content, deliver enhanced antigenic cargo to MIIC vesicles for further processing and presentation. This mechanism may account for the augmented antigen presentation observed in LRBA-KO cells. We have incorporated this interpretation in the revised manuscript in the Discussion section (Lanes: 382 to 385).

2. The interaction between LRBA and FYCO1 is well-documented (Fig. 5), and reduced autophagosome velocity in shLRBA cells is compelling (Fig. 5D-F). However, the manuscript does not connect FYCO1 dysfunction to enhanced antigen presentation. Does FYCO1 knockdown phenocopy LRBA deficiency in MHC-II presentation assays? This could clarify whether impaired autophagosome transport directly contributes to MIIC accumulation.

We thank the reviewer for this comment. While others or we have not shown direct evidence linking FYCO1 dysfunction to enhanced antigen presentation, we agree with the reviewer that FYCO1, due to its role in vesicle trafficking, could potentially affect antigen processing and presentation, and eventually contribute to MIIC vesicles accumulation. This observation has been now included in our Discussion section, lanes: 395-398.

However, we speculate that FYCO1 knockdown may not fully phenocopy LRBA deficiency. This is because autophagosome-lysosome fusion relies not only on FYCO1 but also on the VPS34/VPS15 (PIK3R4) complex, which is also affected in LRBA deficiency. Although both FYCO1 and VPS15 serve distinct molecular functions within the autophagy process, deficiencies in either protein result in overlapping phenotypic outcomes, including autophagosome accumulation, disturbed autophagosome-lysosome fusion, and accumulation of p62 cargo protein (*Jaber N. et al., 2011, PNAS; Parekh V. et al., 2013, J. Immunol.*). Notably, dysfunctional antigen presentation has been described in dendritic cells lacking VPS34 (*Parekh, V. et al., 2017, PNAS*). Thus, we believe that a double knockdown of FYCO1 and VPS15 may be a closer phenocopy of LRBA deficiency than single knockdowns of them. Further studies using single or double FYCO1/VPS15 knockouts in antigen presentation assays will help clarifying the extent to which they truly phenocopy LRBA deficiency in antigen processing and presentation.

Editor:

The referee further commented on the statistical analysis, and this concern should be addressed by being specific in the figure legends about sample size and statistical test used. If the data rely on 2 replicates, please do not perform statistical analysis. In all cases, the individual data points have to be shown in addition to the mean or average, if applicable:

1. Some experiments (e.g., Fig. 2G, n=2) have low sample sizes, and it's unclear whether "n" represents biological or technical replicates. The authors should specify this in the figure legends and, where possible, increase replicates to enhance statistical power.

We thank the reviewer for addressing this concern. Following this recommendation, we have modified each figure legend specifying the biological and technical replicates.

2. Additionally, the use of multiple statistical tests (Mann-Whitney, Welch's t-test, Two-Way ANOVA) without correction for multiple comparisons could inflate type I errors-consider applying a Bonferroni or FDR adjustment."

Following the reviewer's suggestion, we have reevaluated our statistical analysis using corrections for multiple comparisons. This is now indicated in the figure legends.

3. Please also discuss a third concern brought forward, i.e., that the data rely to quite some extent on non-immune cell lines, such as HEK293T and HaCat cells, with only limited evidence provided for autophagy flux and antigen presentation in immune cells, such as LRBA KO B-cells (Figure 4G, S4G-H).

In order to address the reviewer comment, we assessed the degradation of p62, a marker of autophagy flux, under basal conditions and upon autophagy induction with Rapamycin in Ramos B cells. LRBA-KO Ramos B cell lines were generated using the CrisprCas9 approach (Expanded View Figure 7, E and F). **Our flow cytometry analysis revealed diminished p62 degradation following Rapamycin treatment in LRBA-KO Ramos B cells compared to WT cells.** These results confirm our previous observations of p62 accumulation in LRBA-KO HEK293T cells, HaCat cells and shLRBA HeLa cells, further validating the autophagy flux defect in immune cells. This new set of data is now included in Expanded View Figure 4, C and D, and in Results section (Lanes: 203-205).

Specific points:

1. Please supply the movie legends as README.txt file and zip it with the movie. The zip file is uploaded.

Thank you for your suggestion. We have created the movie legends as a README.txt file and have zipped it along with the movie as requested. The zip file has been uploaded for your review.

2. Please provide the exact p-values in the figure legends (unless $p < 0.0001$) and please do not perform statistical analysis based on 2 repeats.

We have revised the figure legends to include the exact p-values where applicable (unless $p < 0.0001$). Additionally, we have removed statistical analyses, which are based on two repeats, as per your suggestion. The updated figure legends are now included in the revised manuscript.

3. Regarding the Author Contributions, we now use CRediT to specify the contributions of each author in the journal submission system. Therefore, please remove the Author Contributions from the manuscript file and make sure that the author contributions in our online manuscript tracking system are correct and up-to-date. The information you specified in the system will be automatically retrieved and typeset into the article. You can enter additional information in the free text box provided, if you wish.

Thank you for the clarification. We have removed the Author Contributions section from the manuscript file as requested. We have also reviewed and updated the author contributions in the online manuscript tracking system.

4. Please update the 'Conflict of interest' paragraph to our new 'Disclosure and competing interests statement'. For more information see:

We have updated the 'Conflict of interest' paragraph to align with the new 'Disclosure and competing interests' statement as per your instructions.

5. The Data Availability statement should only refer to data deposited in public databases.

Please remove the current statement and replace it with: "This study includes no data deposited in external repositories."

We have removed the current Data Availability statement and replaced it with the following, as requested: "This study includes no data deposited in external repositories."

6. References: et al should be used if there are more than 10 authors, i.e., the first 10 authors are listed followed by et al.

Thank you. We have modified our references accordingly.

7. The Expanded View Figure Legends come directly after the main Figure Legends.

In the revised manuscript, we have placed the Expanded View Figure Legends directly after the main Figure Legends, as requested.

8. The nomenclature for Expanded View content is: Movie EV#, Figure EV#.

In the revised manuscript, we have changed the Expanded View nomenclature as indicated.

9. Materials and Methods are called Methods and we need a Reagent and Tools Table (see point 12 below).

This change has been implemented. In addition, we are now providing a Reagent and Tools table according to the instructions.

10. Please provide the Author Checklist (see below).

Author checklist was completed and uploaded along to our revised manuscript.

Dear Dr. Gámez-Díaz

Thank you for the submission of your revised manuscript to EMBO reports and for addressing the remaining concerns from referee #3.

Browsing through the manuscript myself, I noticed a few editorial things that we need before we can proceed with the official acceptance of your study. In particular a few points regarding some of the Western blots shown need to be resolved.

- Please place the Data availability statement before the Acknowledgments.
- Please change "Competing interest" to "Disclosure and Competing Interests Statement".
- Please resolve the following author name discrepancy: Laura-Anne Ligeon in the manuscript vs. Laura-Anne Lingeon in the online submission system.
- The ORCID ID is missing for corresponding author Laura Gamez-Diaz. Please note that you have to link your ORCID account to your account in our manuscript tracking system yourself. Please find instructions here: (<<https://www.embopress.org/page/journal/14693178/authorguide#authorshipguidelines>>)
- Information on funding needs to be part of the Acknowledgments. Please merge the Acknowledgements with the funding section.
- The last two funders and their grant numbers need to be removed from the Comments box in the manuscript tracking system and should be entered as separate funders via More Funders option (Lighthouse Core Facility is funded in part by the Medical Faculty, University of Freiburg (Project Numbers 2021/B3-Fol) and the DFG (Project Number 450392965). [The information in the manuscript tracking system is transferred to production and it is this information that will be used for deposition on our webpage and PubMed (not the one in the manuscript text)].
- We need the two movies as separate ZIP files, i.e., each movie is zipped with its legend and then these two separate ZIP files are uploaded.
- Experiments involving Lrba KO mice: please specify housing conditions and state details of authority granting ethics approval and the reference number.
- Please remove the Reagents and Tools table from the manuscript and upload it as separate file.
- Our data editors noted a few more points in the figure legends that need correction, as follows:
 - Please indicate the statistical test used for data analysis in the legend of figure 1A
 - Please define the error bars in the legend of figure 2G.
 - Please define the scale bar for figure 2F.
- Please remove the Abbreviations section from the manuscript. Abbreviations should be defined in brackets after their first mention in the text, not in a list of abbreviations.
- Materials and Methods should be Methods
- We perform a routine image and data integrity check on all revised manuscript and noticed the following points that need clarification:
 - a) The LRBA blots shown in Figure 1E and Figure EV1C appear identical (IP and input). Please clarify whether these two panels are from the same experiment. Did you perform an IP for Myc-LRBA and detected HA-VPS34 and His-PIK3R4 in the same experiment and blots? Please clarify. If this is indeed the case and you decided to show the blots for VPS34 in the EV figure, this reuse needs to be stated in the figure legends.
 - b) The images shown for the basal control conditions for both, WT and LRBA-KO, appear to be the same in Figure 2E and Figure EV2I. Please clarify whether these control images are appropriate for both panels. Have these experiments been performed at the same time, i.e., one basal conditions and Rapamycin, Torin1, Starvation, and Wortmannin treatment within the same experiment? Please clarify. In case the control is valid for all conditions, please clearly state the reuse in the respective figure legends. More than one control example might be better to show variation and reproducibility.
 - c) The tubulin control blot used in Figure EV2A appears to be the same as that shown in Figure EV6F. EV2A shows data from HEK293T cells, while EV6F appears to come from HaCat cells. Please clarify.

d) The GAPDH blot shown in Figure EV3C appears to be very similar to the tubulin blot in Figure EV7G. LRBA blots appear similar as well. Please check and clarify.

e) Figures 3 and 6 show bad pixelation in cell images. This is most likely produced by the conversion from the 16-bit captured image to the PDF/JPEG image, which can introduce artefacts.

Please provide high-resolution cell images from the original captured 16-bit image.

f) Quantification for Figure 6B: two values for CD4 and CD4+PMA/Iono are the same. Is this correct or a copy-and-paste error? A4 and A5 are the same values and B4 and B5. Please double-check.

- Finally, please provide a summary draft that will accompany the synopsis image:

A) a short (1-2 sentences) summary of the findings and their significance,

B) 2-3 bullet points highlighting key results.

With kind regards,

Point-by-point replies to Editor comments:

Please find below the Reviewers' comments in black and the authors' responses in blue.

Editor:

Browsing through the manuscript myself, I noticed a few editorial things that we need before we can proceed with the official acceptance of your study. In particular, a few points regarding some of the Western blots shown need to be resolved.

1. Please place the Data availability statement before the Acknowledgments.

We have moved the Data Availability statement, as requested.

2. Please change "Competing interest" to "Disclosure and Competing Interests Statement".

The heading has been updated to "Disclosure and Competing Interests Statement", as requested.

3. Please resolve the following author name discrepancy: Laura-Anne Ligeon in the manuscript vs. Laura-Anne Lingeon in the online submission system.

Thank you for bringing this typo to our attention. The correct spelling is Laure-Anne Ligeon, and we have updated the online submission system to match the manuscript.

4. The ORCID ID is missing for corresponding author Laura Gamez-Diaz. Please note that you have to link your ORCID account to your account in our manuscript tracking system yourself. Please find instructions here:

(<<https://www.embopress.org/page/journal/14693178/authorguide#authorshipguidelines>>)

We have successfully linked the ORCID ID of Laura Gámez-Díaz to her account in the tracking system.

5. Information on funding needs to be part of the Acknowledgments. Please merge the Acknowledgements with the funding section.

We have merged these two sections, as requested.

6. The last two funders and their grant numbers need to be removed from the Comments box in the manuscript tracking system and should be entered as separate funders via More Funders option (Lighthouse Core Facility is funded in part by the Medical Faculty, University of Freiburg (Project Numbers 2021/B3-Fol) and the DFG (Project Number 450392965). [The information in the manuscript tracking system is transferred to production and it is this information that will be used for deposition on our webpage and PubMed (not the one in the manuscript text)].

Thank you for the clarification. We have removed the funding details from the comments box and re-entered them as separate funding entries using the More Funders option in the online system, as requested.

7. We need the two movies as separate ZIP files, i.e., each movie is zipped with its legend and then these two separate ZIP files are uploaded.

We have prepared and uploaded two separate ZIP files, each containing one movie along with its corresponding legend, as requested.

8. Experiments involving Lrba KO mice: please specify housing conditions and state details of authority granting ethics approval and the reference number.

Mice were bred and maintained on a C57BL/6 N background under specific pathogen-free (SPF) conditions at the animal facility of the Center for Experimental Models and Transgenic Service (CEMT), University Medical Center Freiburg, Germany. All animal experiments were approved by the local animal ethics committee (Regierungspräsidium Freiburg, Germany) under the following reference numbers: X-15/05F, G-16/19, G-16/94 and G15-168). This information has been added to the Methods section.

9. Please remove the Reagents and Tools table from the manuscript and upload it as separate file.

We have removed the Reagents and Tools table and uploaded it in the system as a separate file.

10. Please indicate the statistical test used for data analysis in the legend of figure 1A.

FuncBase and STRING are databases and web tools that provide functional predictions and pre-calculated protein-protein interaction data, with confidence scores based on integrated evidence (Beaver, J et al., 2010 and Mering et al., 2007). For protein enrichment analysis, both tools use the hypergeometric test, often applying multiple testing correction methods like Benjamin-Hochberg or Bonferroni. As this statistical approach is intrinsic to the platforms and the corresponding references are already cited in the manuscript, we have not included this information in the figure legend. However, we are happy to add it if the editor deems it necessary.

11. Please define the error bars in the legend of figure 2G.

Thank you for noticing, we have added it.

12. Please define the scale bar for figure 2F.

Thank you for noticing, we have added it.

13. Please remove the Abbreviations section from the manuscript. Abbreviations should be defined in brackets after their first mention in the text, not in a list of abbreviations.

The abbreviations section has been removed, and all abbreviations are now defined in brackets at their first appearance.

14. Materials and Methods should be Methods.

The title has been changed to "Methods" as requested.

15. We perform a routine image and data integrity check on all revised manuscript and noticed the following points that need clarification:

a) The LRBA blots shown in Figure 1E and Figure EV1C appear identical (IP and input). Please clarify whether these two panels are from the same experiment. Did you perform an IP for Myc-LRBA and detected HA-VPS34 and His-PIK3R4 in the same experiment and blots? Please clarify. If this is indeed the case and you decided to show the blots for VPS34 in the EV figure, this reuse needs to be stated in the figure legends.

Thank you for your comment, and we apologize for the confusion. Yes, we detected both Myc-VPS34 (using anti-VPS34) and His-PIK3R4 (using anti-PIK3R4) in the same Myc-LRBA IP. As suggested, we have indicated that in the figure legend of Figure 1E and Figure EV1C.

b) The images shown for the basal control conditions for both, WT and LRBA-KO, appear to be the same in Figure 2E and Figure EV2I. Please clarify whether these control images are appropriate for both panels. Have these experiments been performed at the same time, i.e., one basal conditions and Rapamycin, Torin1, Starvation, and Wortmannin treatment within the same experiment? Please clarify. In case the control is valid for all conditions, please clearly state the reuse in the respective figure legends. More than one control example might be better to show variation and reproducibility.

Thank you once again for your comment, and we apologize for the confusion. Yes, all conditions (Basal, Rapamycin, Torin 1, Starvation and Wortmanin) were tested within the same experiment, and the basal control images shown in Figures 2E and EV2I are from the same experimental set. Initially, all these images were part of the same Figure 2E, but during previous revisions, the figure was split into two panels (one for the main figure and the other for the supplementary figure). To avoid confusion and show reproducibility, we have replaced the images in Figure EV2I with additional representative basal control images.

c) The tubulin control blot used in Figure EV2A appears to be the same as that shown in Figure EV6F. EV2A shows data from HEK293T cells, while EV6F appears to come from HaCat cells. Please clarify.

We sincerely apologize for this oversight. Due to an error during figure assembly, the tubulin blot from the HEK293T cells (Figure EV2A) was accidentally also used for the HaCat cells in Figure EV6F. We have now corrected Figure EV6F (HaCat cells) by replacing the duplicated tubulin image with the appropriate blot from the HaCat cells. For transparency, below we show the original, uncropped images of the tubulin blot for both HEK293T cells and HaCat cells.

d) The GAPDH blot shown in Figure EV3C appears to be very similar to the tubulin blot in Figure EV7G. LRBA blots appear similar as well. Please check and clarify.

We apologize for this mistake. Upon review, we found that the labelling in Figure EV7G is incorrect, the blot labelled as Tubulin is actually GAPDH. The images are identical because the Western Blot for LRBA, S6K total, S6K pT389 and GAPDH (Figure EV3C) was performed simultaneously, and the LRBA and GAPDH panels were reused in Figure EV7G. To avoid further confusion, we have changed figure

EV7E by replacing the reused blot with an independent experiment showing the confirmation of shLRBA knockdown in HeLa cells where Tubulin was used as control (new image below).

e) Figures 3 and 6 show bad pixelation in cell images. This is most likely produced by the conversion from the 16-bit captured image to the PDF/JPEG image, which can introduce artefacts. Please provide high-resolution cell images from the original captured 16-bit image.

Thank you for pointing this out. The pixelation issue was indeed caused by the conversion process from the original 16-bit images to PDF during figure preparation. We have now updated the figures with high-resolution images. Unfortunately, for Figure 3, our collaborator could not locate the original 16-bit image of the Torin1+Bafilomycin panels for the LRBA-KO cells. Therefore, we have replaced this panel with another representative images from the same experiment to ensure high resolution. In addition, during this process, we realized that the images in Figure 2F, 2H and 4E were also affected by pixelation due to image conversion. These images have now been re-exported at high resolution. The source data of the new images have also been uploaded to the system.

f) Quantification for Figure 6B: two values for CD4 and CD4+PMA/Iono are the same. Is this correct or a copy-and-paste error? A4 and A5 are the same values and B4 and B5. Please double-check.

Thank you for bringing this to our attention. Upon reviewing our raw data, we confirmed that the identical values were indeed a copy-and-paste error during data transfer. We have now corrected the values in Figure 6B to accurately reflect the original data. The revised Figure 6 and source file have been included in the revised submission.

16. Finally, please provide a summary draft that will accompany the synopsis image:

- A) a short (1-2 sentences) summary of the findings and their significance,
- B) 2-3 bullet points highlighting key results.

This study uncovers a novel role for LRBA in regulating autophagy and antigen presentation, shedding light on the immune dysregulation mechanisms in LRBA deficiency.

- LRBA directly interacts with PIK3R4 and FYCO1, promoting autophagosome formation, transport, and fusion with lysosomes.
- LRBA knockout impairs PI(3)P production and autophagosome-lysosome fusion, leading to accumulation of enlarged autophagosomes with reduced cargo degradation.
- Defective cargo degradation in LRBA-deficient cells increases antigenic peptides in MIIC vesicles, enhancing antigen presentation and T-cell driven proinflammatory cytokine production.

Laura Gámez-Díaz
Institute for Immunodeficiency, Center for Chronic Immunodeficiency (CCI), Medical Center - University of Freiburg
Breisacher Strasse 115
Baden-Württemberg 79106
Germany

Dear Laura,

I am very pleased to accept your manuscript for publication in the next available issue of EMBO reports. Thank you for your contribution to our journal.

Kind regards,

Martina
